# Metagenomic insights into microbial community structure and metabolism in alpine permafrost on the Tibetan Plateau

Luyao Kang[1,2,3], Yutong Song[1,2,3], Rachel Mackelprang [4], Dianye Zhang[1,2], Shuqi Qin[1,2], Leiyi Chen [1,2], Linwei Wu [5], Yunfeng Peng [1,2] & Yuanhe Yang [1,2,3] ✉

Permafrost, characterized by its frozen soil, serves as a unique habitat for diverse microorganisms. Understanding these microbial communities is crucial for predicting the response of permafrost ecosystems to climate change. However, large-scale evidence regarding stratigraphic variations in microbial profiles remains limited. Here, we analyze microbial community structure and functional potential based on 16S *rRNA* gene amplicon sequencing and metagenomic data obtained from an ~1000 km permafrost transect on the Tibetan Plateau. We find that microbial alpha diversity declines but beta diversity increases down the soil profile. Microbial assemblages are primarily governed by dispersal limitation and drift, with the importance of drift decreasing but that of dispersal limitation increasing with soil depth. Moreover, genes related to reduction reactions (e.g., ferric iron reduction, dissimilatory nitrate reduction, and denitrification) are enriched in the subsurface and permafrost layers. In addition, microbial groups involved in alternative electron accepting processes are more diverse and contribute highly to community-level metabolic profiles in the subsurface and permafrost layers, likely reflecting the lower redox potential and more complicated trophic strategies for microorganisms in deeper soils. Overall, these findings provide comprehensive insights into large-scale stratigraphic profiles of microbial community structure and functional potentials in permafrost regions.

Permafrost is defined as any ground that remains continuously frozen at or below 0 °C for a minimum of two consecutive years[1]. Despite the persistent freezing temperatures and oligotrophic conditions, permafrost harbors a pronounced array of microorganisms, which possess the capacity to execute highly intricate metabolic pathways, such as soil organic matter degradation, carbon fixation, methanogenesis, methane oxidation, nitrogen metabolism, and others[1,2]. These diverse microbial metabolisms are critical for biogeochemical processes in permafrost ecosystems, especially in the context of the ongoing permafrost thawing under climate warming[3]. The acceleration of microbial-mediated biogeochemical cycles may promote the emission of greenhouse gases (e.g., $CO_2$, $CH_4$, and $N_2O$), ultimately leading to both carbon-climate and non-carbon-climate feedbacks in permafrost regions[3,4]. Recognizing the crucial role of microorganisms in governing biogeochemical processes in permafrost ecosystem, and deciphering their compositional and functional profiles can enhance our

---

[1]State Key Laboratory of Vegetation and Environmental Change, Institute of Botany, Chinese Academy of Sciences, Beijing, China. [2]China National Botanical Garden, Beijing, China. [3]University of Chinese Academy of Sciences, Beijing, China. [4]California State University Northridge, Northridge, CA, USA. [5]Institute of Ecology, Key Laboratory for Earth Surface Processes of the Ministry of Education, College of Urban and Environmental Sciences, Peking University, Beijing, China. ✉e-mail: yhyang@ibcas.ac.cn

comprehension of how biogeochemical processes will respond to climate changes in this globally important ecosystem[3].

Due to the crucial role of microorganisms in mediating the biogeochemical processes in permafrost regions, recent research in these areas has concentrated on revealing the microbial profiles[3]. However, given that these studies have primarily addressed permafrost microorganisms at the site-specific level, mainly focusing on the active layer (0–50 cm)[5–8], there still lacks a systematic profiling of the large-scale stratigraphic characteristics of microorganisms in permafrost regions. In fact, microbial biogeographic patterns and assemblage mechanisms may differ among soil layers due to lateral and vertical variations in water phase, osmotic potential, and nutrient availability across permafrost regions[1]. Specifically, recurrent freeze-thaw cycles in the active layer act as disruptive events that directly induce fluctuations in microbial population sizes via demographic processes[9,10], which may consequently lead to prominent effects of ecological drift on the microbial assemblages. Additionally, soil structure, moisture, and substrate availability vary with the freeze-thaw cycles[11,12], potentially leading to niche differentiation (selection) due to the difference in microbial physiologic traits[12]. Compared with the active layer, the frozen permafrost soils have a higher ice content, which acts as a physical barrier that impedes microbial migration due to the lack of continuous water pathways[13,14], potentially resulting in pronounced dispersal limitation. Meanwhile, niche selection mediated by harsh conditions (e.g., oxygen depletion, nutrient scarcity, or freezing temperature) may reduce microbial diversity in permafrost soils during prolonged frozen periods, while speciation via mutation may increase it[1,15]. However, the strength of both niche selection and speciation may be muted in the permafrost layer by lower metabolic activity or a resistant survival strategy, such as dormancy[16]. Overall, the interplay of the aforementioned ecological processes can induce distinct biogeographic patterns among the soil layers, but our understanding on this issue is still in its infancy[15].

Besides the community assemblage, microbial metabolic potential could also vary over the vertical soil profile. One notable characteristic of deep permafrost soils is that the microbial community structures are affected by the redox status due to the limited oxygen availability[17,18]. In such conditions, microorganisms are more likely to engage in anaerobic respiration and fermentation for soil organic matter degradation[19]. These processes necessitate a series of reductive reactions involving diverse alternative electron acceptors, such as nitrate, sulfate, ferric iron, carbon dioxide, and small organic molecules[2]. Therefore, genes associated with reduction reactions pertinent to the aforementioned elements may become more prevalent, and the corresponding taxa involved in these reduction pathways may be more diverse, showing a greater contribution to the community-level metabolic profiles with soil depth. Alternatively, microorganisms may also be less reliant on alternative electron acceptors due to the slower microbial activities in deeper soils[20]. Despite the recognitions of these microbial processes, there exists a dearth of knowledge concerning their variability across different soil strata, and that of their corresponding functional groups, as well as their respective contributions to the overall community metabolism across permafrost regions.

To fill the aforementioned knowledge gaps, we established 22 sampling sites along an -1000 km permafrost transect on the Tibetan Plateau (Fig. 1a; Supplementary Table 1), the largest permafrost region outside the high latitudes[21]. We collected soil samples from the 0–10 cm and, 30–50 cm layers, and from the uppermost 50 cm thick permafrost layer, representing the surface, subsurface, and permafrost layers of the soil profile, respectively. We employed both amplicon and metagenomic sequencing methods to examine the microbial community structure and functional profiles. Additionally, we retrieved climatic, plant, and anthropogenic factors, and determined environmental variables, including substrate properties and edaphic factors, to explore the environmental effects on structuring the microbial communities. With these measurements, we aimed to test the following two hypotheses: (1) microbial assembly mechanisms differ among soil layers, with the contribution of stochastic drift decreasing, but that of dispersal limitation increasing with soil depth. (2) The relative abundance of genes participating in reduction reactions related to alternative electron acceptors is higher, and the taxa engaged in the redox reactions become more diverse with increasing soil depth. Consistent with the hypotheses, our results show that microbial alpha diversity declines, while beta diversity increases, down the soil profile. Microbial assemblages are primarily governed by dispersal limitation and drift. The importance of drift decreases from the surface active layer to the deep permafrost deposits, while that of dispersal limitation increases. In the subsurface and permafrost layers, functional genes related to reduction reactions are enriched, and the corresponding taxa participating in redox reactions are more diverse and contribute highly to community-level metabolic profiles. These findings lay the groundwork for a comprehensive understanding of microbial biogeographic patterns, assembly mechanisms, and metabolic attributes in high-altitude permafrost regions.

## Results and discussion

### Decreasing microbial alpha, but increasing beta diversity with soil depth

To discern disparities in microbial diversity among soil strata, we first determined the alpha diversity indices, including the Shannon-Wiener index and Faith's index, and beta diversity metrics such as the Bray-Curtis distance and β-mean nearest taxon distance based on the amplicon data. Our results showed that microbial alpha diversity declined, while beta diversity (spatial variations) increased with soil depth (Fig. 1b–e), which is consistent with our first hypothesis. The lower alpha diversity in permafrost deposits could be due to permafrost habitats being typically oligotrophic, having temperatures below freezing, and low water availability. Such harsh conditions may impose higher selective pressure on microorganisms living in permafrost deposits than those in the active layer, and ultimately lead to lower diversity[1,22]. In contrast to alpha diversity, we observed increasing spatial variability (beta diversity) of microorganisms with soil depth (Fig. 1d–e), indicating that taxonomic heterogenization and phylogenetic divergence were occurring within the permafrost soils. Generally, microorganisms in the active layer can transport via wind and pore water channels, thereby increasing species migration among communities and leading to a more homogeneous community composition[23]. In contrast to those in the active layer, microbes in permafrost deposits are believed to have been entrained during permafrost formation by microbial taxa from ancient paleoenvironments[2,24,25]. Over their prolonged evolution within the permafrost, microbial communities are highly isolated, particularly as a result of the soil freezing which acts as a physical barrier[26]. Consequently, microbial communities at different sites will diverge from one another, resulting in significant spatial variations.

Multivariable statistics utilizing Bray-Curtis dissimilarity matrices unveiled a notable differentiation in taxonomic compositions among different soil layers (Supplementary Table 2). Specifically, we identified a total of 12,855 amplicon sequence variants (ASVs), primarily belonging to the phyla Proteobacteria, Actinobacteria, Chloroflexi, and Acidobacteria (Supplementary Fig. 1). Of these ASVs, 9850 (76.6%) shared among the three soil layers, with the permafrost layer harboring the highest number of unique species (269 ASVs), followed by the surface layer (187 ASVs) (Fig. 1f). Microbes in the surface soils were dominated by Proteobacteria, whereas Actinobacteria displayed an elevated relative abundance within the subsurface and permafrost layers (Fig. 1g; Supplementary Fig. 1). Such a pattern can be ascribed to the intrinsic attributes of these two microbial groups. It has been suggested that the higher abundance of Proteobacteria in the surface

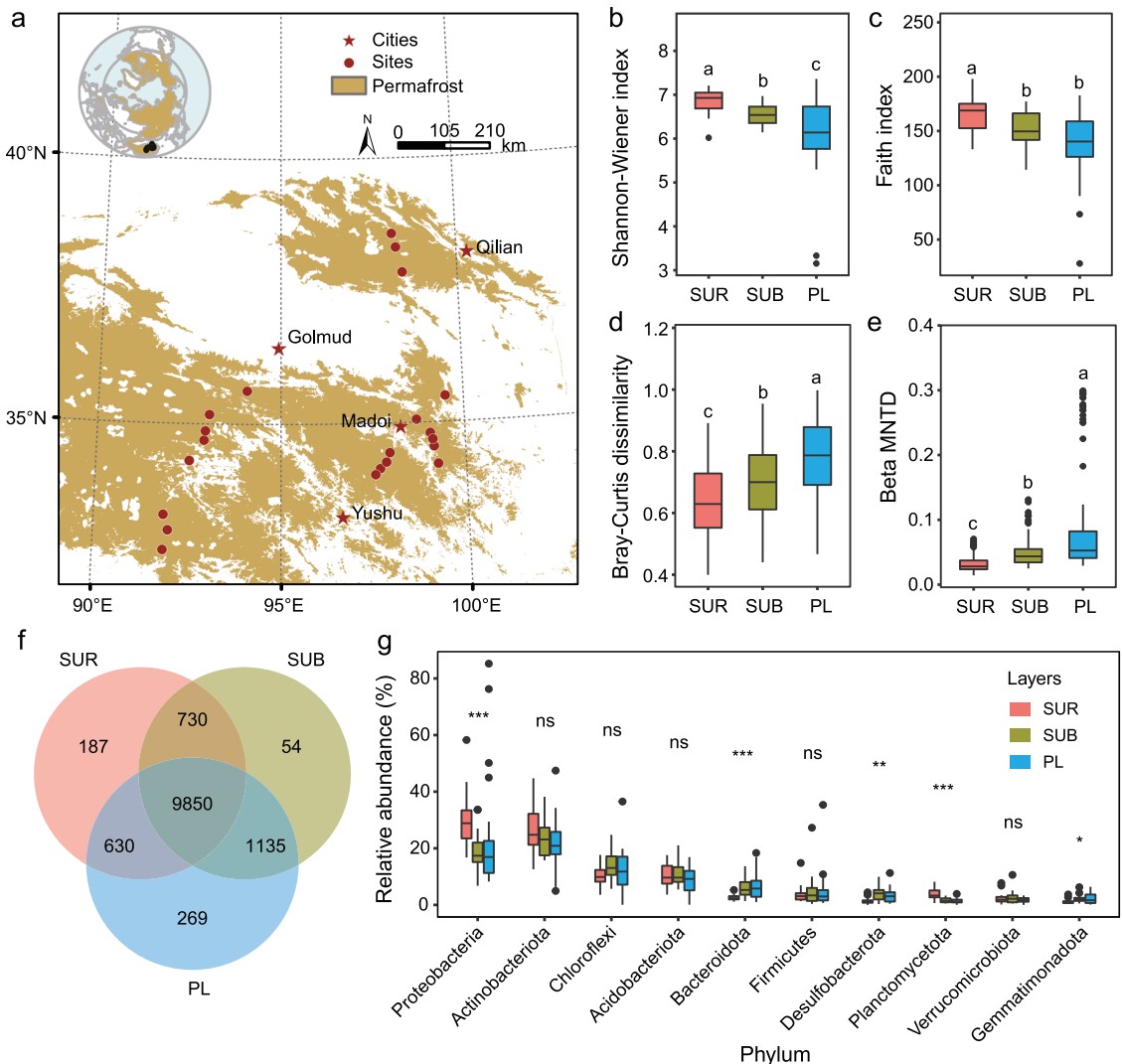

**Fig. 1 | Distribution of sampling sites and comparisons of microbial diversity and composition among various soil layers. a** Geographic distribution of the sampling sites across the Tibetan alpine permafrost region. Source of the spatial distribution of permafrost across the Northern Hemisphere is provided by the National Snow & Ice Data Center[103], while that of the Tibetan Plateau is obtained from ref. 61. The map was drawn by ArcMap 10.6 (Environmental Systems Research Institute, Inc., Redlands, CA, USA). Differences in microbial alpha diversity consisting of (**b**) taxonomic diversity (Shannon-Wiener index) ($n = 22$) and (**c**) phylogenetic diversity (Faith index) ($n = 22$) among three soil layers. Differences in microbial beta diversity consisting of (**d**) taxonomic variation (determined by pair Bray-Curtis distance) ($n = 231$) and (**e**) beta mean nearest taxon distance (Beta MNTD) ($n = 231$) among three soil layers. **f** The number of shared and unique ASVs among three soil layers. **g** Microbial composition and their differences among soil layers, only the top 10 phyla are shown here ($n = 22$). Different lowercase letters in box plots denote significant differences for the variables with soil depth (determined by two-sided pairwise Wilcoxon test, $P < 0.05$). *$P < 0.05$, **$P < 0.01$, ***$P < 0.001$, and ns: non-significant. SUR, surface (0–10 cm); SUB, subsurface (30–50 cm); PL, permafrost layer–the uppermost 50 cm thick layer of permafrost soil. Central line and whiskers in each box represent the median and 1.5 times the interquartile range, respectively. Boxes indicate the interquartile range between 25th and 75th percentile. Single points are outliers. Source data are provided as a Source Data file.

layer may be linked to elevated levels of carbon and nutrients[27]. For the Actinobacteria, they are distinguished by an expansive repertoire of secondary metabolites and these attributes could enhance their resistance to selective pressure, thereby promoting their survival in permafrost deposits[27,28]. The occupancy (the relative frequency of a given species occurs within all samples) and specificity (the average abundance of a given species within all samples) analysis[29] revealed that the numbers of specialist species in surface, subsurface, and permafrost layers were 181, 27, and 30, respectively (Fig. 2a). Specialist species in the surface layer were mostly α-Proteobacteria, γ-Proteobacteria, and Actinobacteria (Fig. 2b), but Bacteroidia, γ-Proteobacteria and Actinobacteriota were found to be both specific and common in the subsurface layer (Fig. 2c). Interestingly, the Thermoleophilia organisms (i.e., Solirubrobacterales and Gaiellales), belonging to the phylum Actinobacteria, were specialist species in the

permafrost layer (Fig. 2d). Thermoleophilia are generally regarded as thermophilic taxa, frequently found to inhabit geothermal hot springs[30]. However, several studies have reported that Thermoleophilia organisms are also cold-adapted and thrive in cold extreme environments, such as Antarctic soil and permafrost soil in Alaska[7,31]. Their higher habitat occupancy in extreme environments might be attributed to specific features, including being strict chemoorganotrophs, halophilic, and possessing DNA repair mechanisms[32,33], which may enhance the adaptability of these microorganisms to extreme environments.

**Microbial communities are more shaped by dispersal limitation in the permafrost layer**
Based on the partial Mantel test and null model analysis, we explored the differences in the microbial community assembly mechanisms

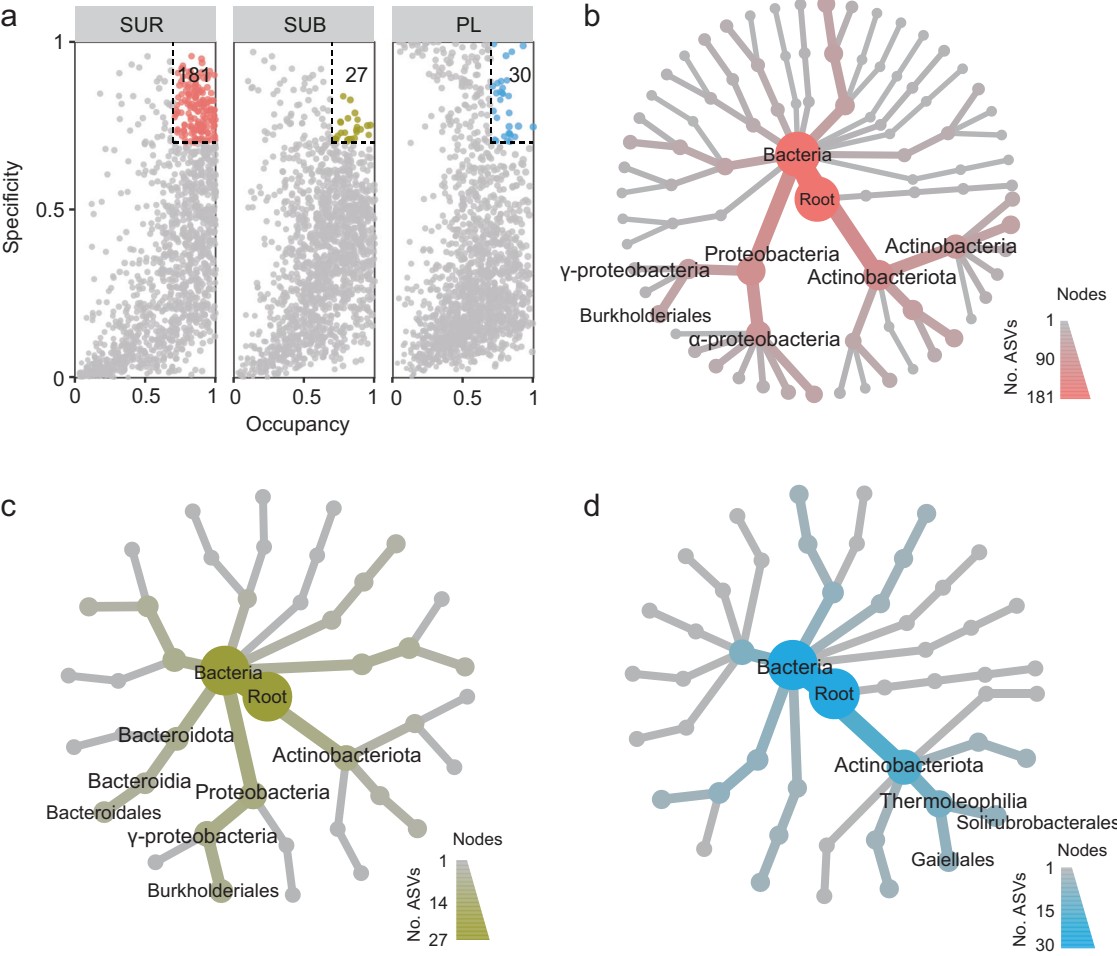

**Fig. 2 | The profile of specialist species in three soil layers. a** The specificity-occupancy plot shows the distribution and specificity of the abundant amplicon sequence variants (ASVs) with a mean relative abundance higher than 0.01% in each layer. ASVs with specificity and occupancy greater or equal to 0.7 are specialist species[95]. Taxonomic trees plotted by the metacoder package[104] illustrate the number and composition of specialist species in (**b**) surface layer (SUR), (**c**) subsurface layer (SUB), and (**d**) permafrost layer (PL). Dark-colored taxa have a higher number of ASVs within the given taxon in the soil layer. Source data are provided as a Source Data file.

across soil layers. We first reduced the variable redundancy by using cluster analysis, and thirteen variables [e.g., mean annual air temperature (MAT), soil pH, and clay content] were retained for the partial Mantel test (Supplementary Fig. 2). Most of the selected environmental variables exhibited significant correlations with microbial compositional variations, with pH and the aridity index (AI: determined by dividing mean annual precipitation by mean annual potential evapotranspiration[34]) emerging as the primary predictors of microbial assemblages (Fig. 3a). Specifically, the microbial community structure in the surface layer was most significantly associated with pH and AI, followed by Normalized Difference Vegetation Index (NDVI), plant species richness, clay content, soil moisture, and the labile carbon pool I (listed in order of decreasing partial Mantel's *r*) (Fig. 3a). In the subsurface layer, the microbial community structure exhibited significant correlations with pH and AI, followed by soil moisture and plant species richness. However, no environmental variables displayed significant associations with microbial community structure in the permafrost layer (Fig. 3a). Generally, soil pH and AI are considered to be good predictors for discerning soil microbial community structure in surface soils[35,36]. They wield the capacity to exert environmental selection over the microbial structure by altering resource availability, energy-related processes, and the prevalence of inhibitory substances[36–38]. In spite of the detected importance of soil pH and AI, the effects of other variables on microbial communities should not be

neglected. For example, soil redox status was observed to exert a prominent effect on community composition and diversity in tundra soils[18]. Therefore, future studies are encouraged to include more explanatory environmental variables, such as ferric irons ($Fe^{3+}$) and electrical conductivity, to further advance our understanding of the underlying mechanisms for microbial communities in permafrost ecosystems. Overall, these results emphasize the crucial roles of environmental selection in shaping the microbial communities in Tibetan alpine permafrost region.

Despite the significant effects of environmental factors (selection) on microbial community structure, the null model analysis produced compelling evidence that two stochastic ecological processes [i.e., dispersal limitation, drift (and others)] predominantly governed the microbial communities (Fig. 3b), but their relative importance varied with soil depth (Fig. 3c). Specifically, drift (and others) had the highest relative importance in shaping the microbial assemblages, followed very closely by dispersal limitation (Fig. 3b). For both the subsurface and permafrost layers, dispersal limitation was identified as making the greatest contribution to structuring microbial communities, followed by the drift (and others) (Fig. 3b). Further results showed that the relative contribution of dispersal limitation was higher in permafrost deposits than in the other two soil layers, while the importance of drift (and others) significantly decreased with soil depth (Fig. 3c). The disparity in the contribution of these ecological

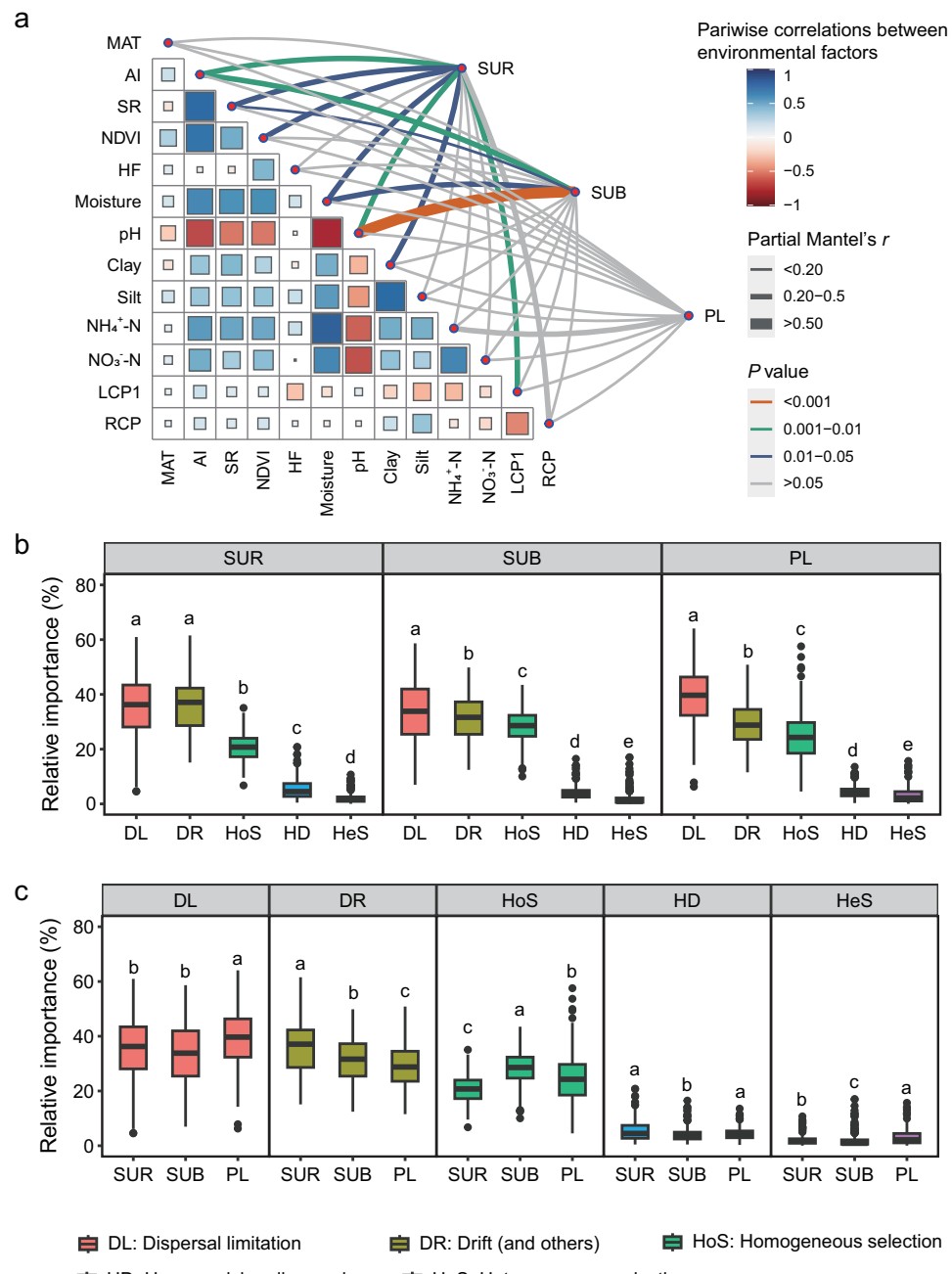

**Fig. 3 | Environmental factors and ecological processes shaping microbial community structure. a** Associations of the microbial community structure (determined by Bray-Curtis distance) with environmental factors (determined by Euclidean distance) using the partial Mantel test. Partial Mantel's *r* values are indicated by the edge width, while the statistical significance is denoted by the edge color. Pairwise correlations of environmental variables are depicted with a color gradient reflecting Spearman's correlation coefficient. **b** The relative contribution of each ecological process driving microbial community assembly within the layer based on null model analysis (*n* = 231). **c** The difference in the relative importance of ecological processes among three soil layers (*n* = 231). Different lowercase letters in

box plots indicate significant differences for the ecological processes with soil depth (determined by two-sided Wilcoxon test, *P* < 0.05). SUR surface layer, SUB subsurface layer, PL permafrost layer, MAT mean annual air temperature, AI aridity index, NDVI Normalized Difference Vegetation Index, SR plant species richness, HF human footprint index, LCP1 labile carbon pool I (mainly polysaccharides), RCP recalcitrant carbon pool. Central line and whiskers in each box represent the median and 1.5 times the interquartile range, respectively. Boxes indicate the interquartile range between 25th and 75th percentile. Single points are outliers. Source data are provided as a Source Data file.

processes among the soil layers may be attributed to differences in habitat conditions. Diurnal and seasonal freeze-thaw cycles act as perturbations that induce periodic declines in microbial populations, thus giving rise to discernible fluctuations in population size and instigating a conspicuous drift effect[39]. Given that the frequency of the freeze-thaw cycles declines with soil depth (Supplementary Fig. 3), the effects of drift (and others) in shaping microbial structure

decreases from the surface layer to the deep soil layers. Conversely, since permafrost deposits are frozen, with limited water availability, the lack of continuous water pathways impedes microbial mobility and ultimately engenders a more pronounced dispersal limitation[40]. This intensified dispersal limitation may be posited as a contributory factor to the higher spatial variability observed within the permafrost layer (Fig. 1d-e). Besides stochastic processes, deterministic processes,

particularly the homogeneous selection, also played an important role in structuring the microbial communities. This process had its highest relative importance in the subsurface layers (Fig. 3c). Such a pattern may reflect the situation in which subsoils usually have relatively uniform environmental conditions. Species adapted to these specific conditions are more likely to thrive, consequently leading to homogeneous selection[41].

## Enriched functional genes involved in reduction reactions in deeper layers

To discern the microbial functional profiles, we annotated metagenome reads by comparing them to the Kyoto Encyclopedia of Genes and Genomes (KEGG) (http://www.kegg.jp) database. Our results revealed that functional genes were mainly involved in carbohydrate, amino acid, and energy metabolism (Supplementary Fig. 4a). Functional compositions differed significantly with soil depth and were more analogous in the two deeper soil layers (i.e., subsurface and permafrost layers) (Supplementary Table 2; Supplementary Fig. 4b). The functional gene composition in the permafrost layer exhibited higher spatial variability than in the other two layers (Supplementary Fig. 4c), and their relative abundance varied among soil layers (Fig. 4). In particular, we observed a higher abundance of genes related to assimilatory nitrate reduction (*nasB*, *NIT-6*, and *nirA*), nitrogen fixation (*nifDKH*), and organic nitrogen metabolism (*gdh*, *glsA*, and *ureAC*) in the surface layer (Fig. 4). The elevation of these genes suggested that microorganisms may have higher demand for nitrogen in the topsoil, which may be induced by higher nitrogen limitation[42] and more intense competition for N between plants and soil microorganisms[43]. We also found that genes participating in assimilatory sulfate reduction (*cysDNC*) and sulfide oxidation (*sqr* and *fccAB*) enriched in surface soil (Fig. 4), indicating the higher demand for reduced sulfur for the formation of amino acids and energy generation.

In contrast to the surface layer, we observed an elevation in the abundance of genes linked to the degradation of hemicellulose, cellulose, and pectin in the subsurface and permafrost layers (Fig. 4a). These genes are regarded as important in mediating the responses of permafrost carbon cycle to climate warming on the Tibetan Plateau[44]. The majority of genes involved in fermentation processes, such as pyruvate oxidation, pyruvate formate lyase, and acetogenesis, showed an increase in abundance in the deeper soils (Fig. 4a). Other genes related to reduction reactions, including dissimilatory nitrate reduction (*NarGHI* and *napB*), denitrification (*nirS* and *nosZ*), polysulphide reduction (*sreB*, *gydABDG*, and *psrAC*), sulfite reduction (*dsrAB* and *asrABC*), tetrathionate reduction (*ttrABC*), $Fe^{3+}$ reduction (*MtrCAB*), $SeO_4^{2-}$ reduction (*ygfMK* and *xdhD*), and methanogenesis (*mcrA*), were also found to increase with soil depth (Fig. 4b–d). The augmentation of fermentation and reduction in inorganic compounds reflected the fact that microbes had adapted to thrive in deeper soils via anaerobic metabolic pathways. Such adaptation is probably due to redox conditions that are highly selective for species that can survive under anoxic conditions for prolonged periods. Generally, the utilization of organic carbon by microorganisms necessitates the availability of terminal electron acceptors, with the preferred order being $O_2$, $NO_3^-$, $Fe^{3+}$, and $SO_4^{2-}$, followed by methanogenesis and other small organic molecules[1,2]. Given that both the subsurface and permafrost layer are more anoxic than the surface layer, microorganisms colonized in these layers tend to utilize the alternative electron acceptors (e.g., $NO_3^-$, $Fe^{3+}$, and $SO_4^{2-}$) to facilitate the anaerobic degradation process[2,19].

## Taxa are diversely engaged in the redox reactions in deeper layers

To decipher the differences in metabolic profiles across the soil layers, we binned the metagenomic reads into metagenomic-assembled genomes (MAGs) and explored their functional capabilities in biogeochemical processes. In total, we obtained 274 medium-quality MAGs (>70% completeness and <10% contamination) (Fig. 5a, Supplementary Data 1). Most of the genomes were annotated as Actinobacteria (e.g., Thermoleophilia, UBA4738), Acidobacteria, or Proteobacteria (Fig. 5a; Supplementary Fig. 5). Most of the genomes (273 MAGs) possessed a capacity for iron reduction (Fig. 5b; Supplementary Note 1), implying that iron reduction contributed highly to organic material oxidation by serving as the terminal electron acceptor during microbial anaerobic respiration[19]. The amino acids utilization process was also ubiquitously among the genomes (Fig. 5b; Supplementary Note 1), indicating that microorganisms were compelled to utilize less energetically favorable substrates, possibly due to the inaccessibility of carbohydrate-rich organic materials[25]. The prevalence of other pathways, including acetogenesis, acyl-CoA dehydrogenase, acetate to acetyl-CoA, chitin degrading, and sulfur oxidation, suggested exceptional metabolic versatility of the microbial groups in the permafrost ecosystem (Fig. 5b; Supplementary Note 1). Furthermore, principal coordinate analysis revealed a notable separation of microbial communities among the three soil layers, with the communities in the permafrost layer being more similar to those in the subsurface layer (Fig. 5c; Supplementary Table 2). Additionally, we observed a diminishment in average genome size down the soil profile (Fig. 5d), indicating that microorganisms inhabiting permafrost environments possessed relatively compact and simplified genomes. This finding suggests that microorganisms may employ the genome reduction strategy that aims at mitigating the metabolic expenditure entailed in DNA replication, enhancing their fitness in the face of the nutrient-scarce environment of permafrost deposits[45,46]. It should be noted that the difference in the recovery rate of MAGs during the binning process (Supplementary Fig. 6) may also lead to variation in the estimation of genome size[47], which could, potentially, induce the decreasing pattern observed in this study.

To further evaluate the importance of metabolic pathways and the functional capacity of each microbial taxa at the community level, we calculated community-level metabolic weight scores (MW-scores) for each functional pathway, along with the percentage contribution of each microbial phylum to these scores[48]. A higher MW-score indicates a larger contribution of a specific pathway to the community-level metabolic profiles and vice versa[48]. Our results showed that both heterotrophic pathways (e.g., amino acid utilization, fermentation, complex carbon degradation, and fatty acid degradation) and autotrophic processes (e.g., CO oxidation, sulfur oxidation, and hydrogen oxidation) exhibited important contributions to the microbial community-level metabolic profiles (Fig. 6a; Supplementary Note 2), demonstrating the intricacies of microbial energy conservation[49,50]. The percentage contribution of microbial taxa showed that the metabolic contributions of the microorganisms in the surface soil were chiefly accounted for by α-Proteobacteria and Actinobacteria (Fig. 6b). The higher percentage contribution of α-Proteobacteria in surface soils may be ascribed to the higher soil carbon and nutrition content in surface soil (Supplementary Fig. 7), which provide more favorable substrate conditions that match the growth requirements of these taxa[51,52]. These observations align with our findings of the heightened metabolic contributions of α-Proteobacteria in carbon decomposition (Fig. 6a; Supplementary Note 3). Additionally, Actinobacteria made a substantial contribution to the microbial metabolic activities in the surface layer (Fig. 6b). Actinobacteria were ubiquitous in the extremobiosphere and involved in a variety of functional processes, including complex carbon degradation, nitrogen cycling, and stress response[5]. Their extraordinary aptitude for producing a wide range of enzymes and secondary metabolites enables them to flourish and persevere in the face of arduous environmental stress[53]. In line with this argument, we also observed notable contributions of Actinobacteria to diverse metabolic pathways, including most of the complex carbon oxidation and the redox reactions involved in the nitrogen, sulfur, and iron metabolic pathways (Fig. 6a; Supplementary Note 3).

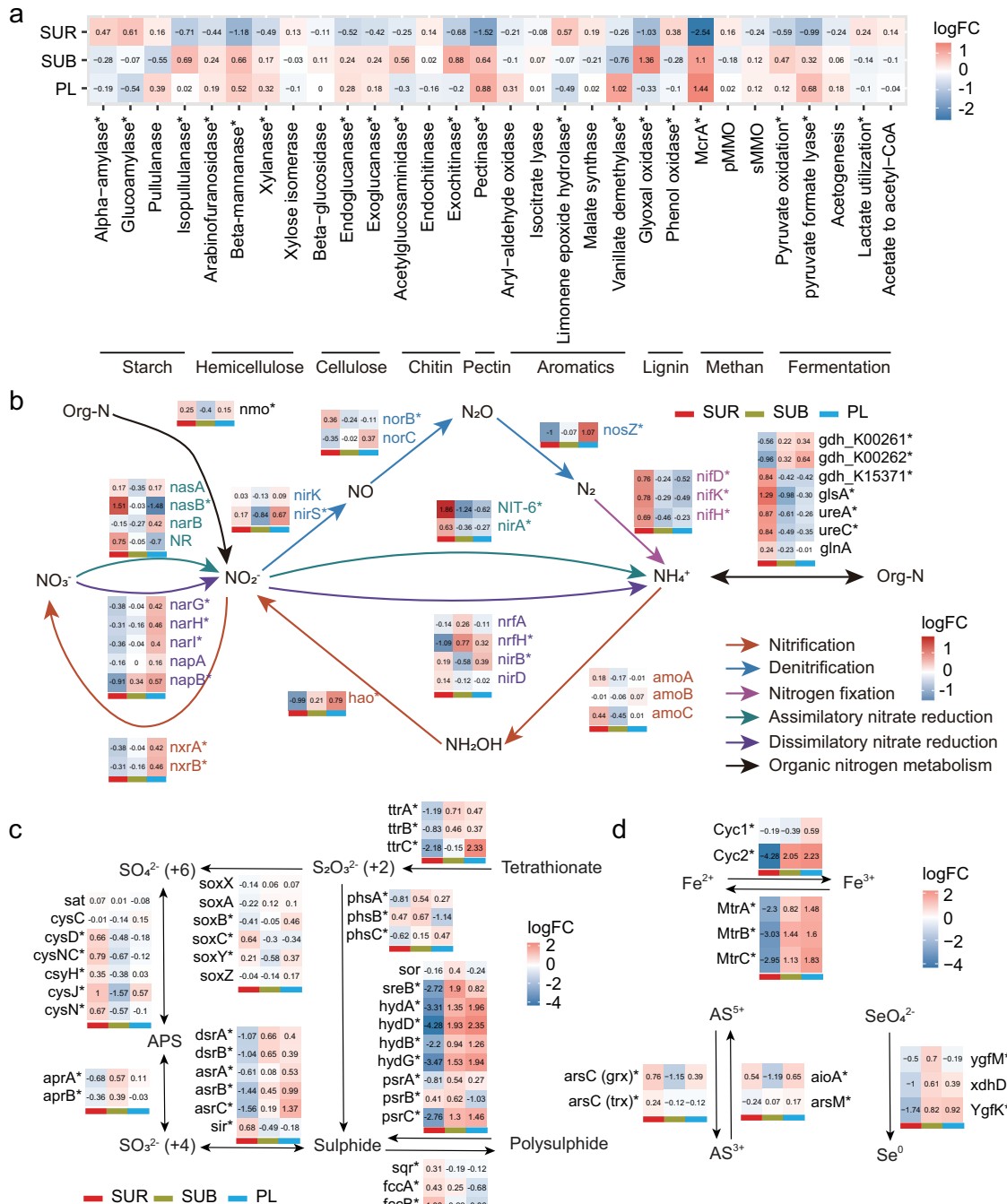

**Fig. 4 | Depth variations of functional potentials of microbiomes in alpine permafrost on the Tibetan Plateau.** Heatmap showing the enrichment of functional genes involved in (**a**) carbon degradation, (**b**) nitrogen cycling, (**c**) sulfur cycling, and (**d**) other elements such as iron, arsenic, and selenium metabolic pathways among three soil layers. The significance of the changes in gene abundance are evaluated by generalized linear model with a negative binomial distribution using edgeR[100] package. $P$ values were obtained from two-sided Likelihood Ratio Tests (LRTs) and adjusted for multiple comparisons using the Benjamini-Hochberg false discovery rate (FDR) procedure. Genes with significant changes in abundance ($P < 0.05$) are marked with an asterisk (*). LogFC: log2-fold change. The full names of the genes in this figure are listed in Supplementary Data 2. SUR surface layer, SUB subsurface layer, PL permafrost layer. Source data are provided as a Source Data file.

In contrast to the surface layer observations, we found that not only Actinobacteria but also more diverse taxa such as Desulfobacterota, γ-Proteobacteria, Chloroflexota, Acidobacteriota, and Methylomirabilota, occupied a significant functional fraction related to the redox reactions in the subsoils and permafrost soils (Fig. 6b; Supplementary Note 3). Given that soil conditions become more anoxic with depth, microbial anaerobic respiration and fermentation are essential pathways for the anaerobic decomposition of soil organic matter[19,54]. The majority of identified anaerobic respiration genes, including those for nitrate reduction (*narGH*), sulfate reduction, and iron reduction, were prevalent in taxa within Actinobacteriota (as shown by the higher contribution on the metabolic weight scores in Fig. 6a). Likewise, genes encoding fermentation including pyruvate oxidation, pyruvate formate lyase, and acetogenesis were abundant in Actinobacteriota and several other taxa (e.g., Acidobacteriota and Proteobacteria) (Fig. 6a, Supplementary Fig. 8). This evidence

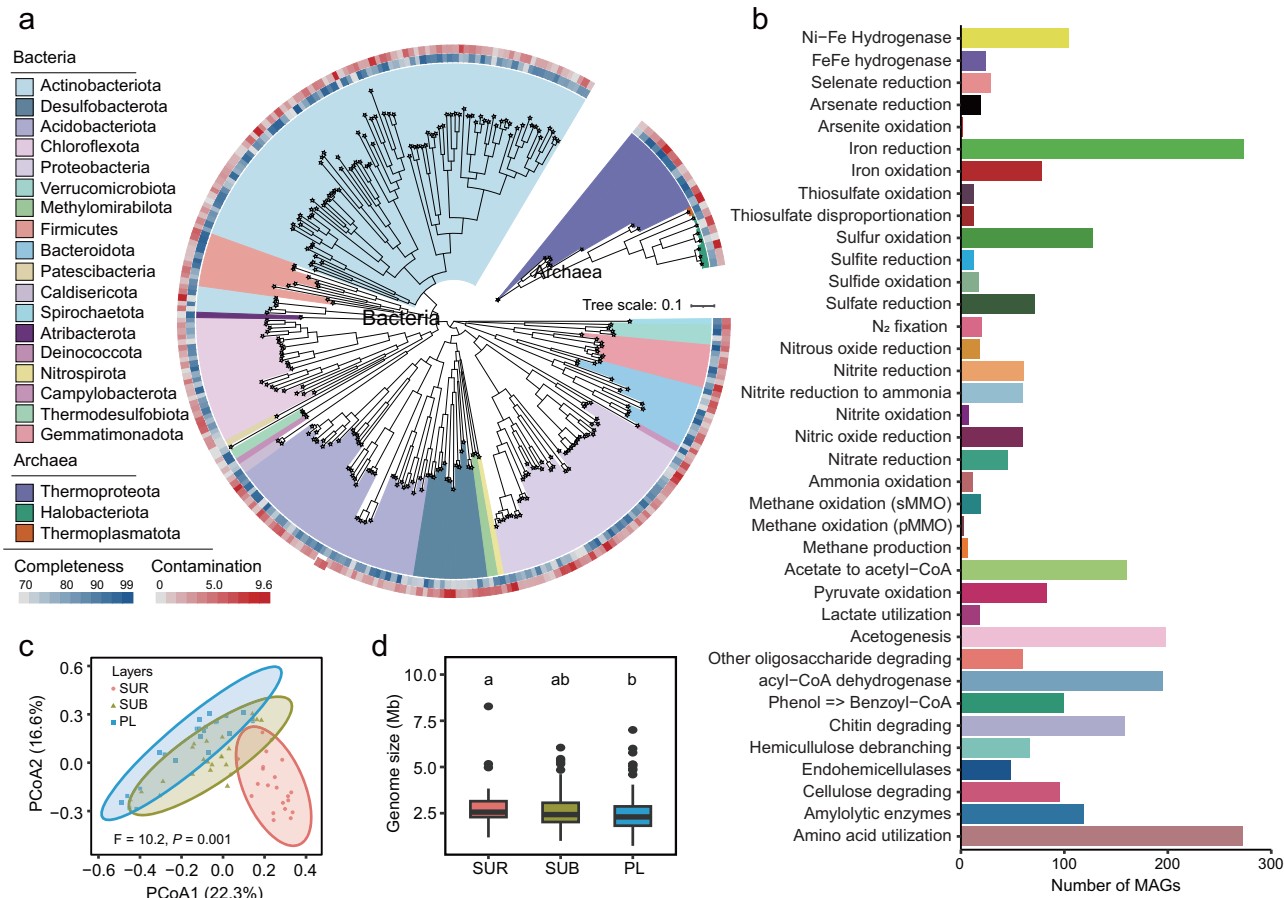

**Fig. 5 | Profile of the phylogeny, taxonomy, metabolic pathways, and genome size of metagenome-assembled genomes (MAGs). a** Maximum-likelihood phylogenetic tree of the 274 MAGs from GTDB, and the completeness and contamination of MAGs. **b** The number of genomes per metabolic pathway. **c** Principal coordinate analysis (PCoA) showing the difference of the relative abundance and composition of MAGs among three soil layers based on Bray-Curtis dissimilarity index ($n = 22$). The significant difference of the MAGs among three soil layers is tested by the permutational multivariate analysis of variance. **d** Comparison of estimated genome size of all MAGs found in the surface (SUR: $n = 34$), subsurface (SUB: $n = 94$), and permafrost (PL: $n = 146$) layers. $P$ values were estimated using two-sided Wilcoxon test. Different lowercase letters in box plot denote significant differences for the genome size with soil depth ($P < 0.05$). Central line and whiskers in each box represent the median and 1.5 times the interquartile range, respectively. Boxes indicate the interquartile range between 25th and 75th percentile. Single points are outliers. Source data are provided as a Source Data file.

illustrated the versatile metabolic potential of Actinobacteria, implying the critical role of Actinobacteria in mediating biogeochemical processes in permafrost ecosystems. Additionally, other taxa had adapted to thrive in specific redox niches within the electron transfer reactions. For instance, Desulfobacterota appeared to be the primary sulfite reducers, also played important roles in thiosulfate disproportionation and nitrite reduction (*octR*) (Fig. 6a). Populations belonging to γ-Proteobacteria were essential contributors to oxidation processes (including Nitrite ammonification, Sulfide oxidation, Thiosulfate oxidation, Iron oxidation, and Arsenite oxidation) (Fig. 6a, Supplementary Note 3). The Chloroflexi, are reported to have many fermentative members, which play crucial roles in anaerobic carbon degradation in permafrost[55]. However, in this study, we observed only a moderate contribution of Chloroflexi to fermentation, and these taxa appeared to be important to the nitrite reduction process (*nirKS* and *octR*) (Fig. 6a). Moreover, taxa belonging to Methylomirabilota are characterized for their capacity to couple anaerobic methane oxidation with nitrite reduction in anoxic environments[56], which allows them to adapt better in anaerobic environments. Likewise, Acidobacteria made large contributions to arsenate reduction, selenate reduction, nitrate reduction (*napAB*), thiosulfate disproportionation, and iron oxidation (Fig. 6a). Such diverse redox metabolism may enable their survival in anoxia and nutrient-poor conditions, consequently allowing them to

be widely present in global soils[57]. These various taxa, which have adapted to thrive in specific redox niches, ultimately co-dominant the biogeochemical processes in deeper soils[18]. Collectively, these results highlighted the diverse microbial species engaged in redox reactions and the more complicated trophic strategies for microorganisms in subsoils and permafrost deposits.

Although the present study revealed the large-scale stratigraphic profile of microbial community structure and functional potentials in Tibetan alpine permafrost regions, there are three limitations that need to be addressed. First, our experiments rely on DNA-based metagenomic techniques that cannot provide insight into which genes are actively expressed and what specific biochemical functions are being carried out at a given time[2,58]. Therefore, there may be a disconnection between the functional potentials we measured and the actual functions being realized. Second, ~40% of the DNA was found to be extracellular or originated from cells that were no longer intact[59]. Constant subzero temperatures provide ideal preservation conditions for inactive or dead cells in permafrost deposits, potentially leading to an even higher proportion of relic DNA[1]. This could introduce some bias into our assessments of microbial structure and function, although some studies have suggested that removing relic DNA does not substantially impact microbial community structure in permafrost[60]. Consequently, the extent to which relic DNA, originating

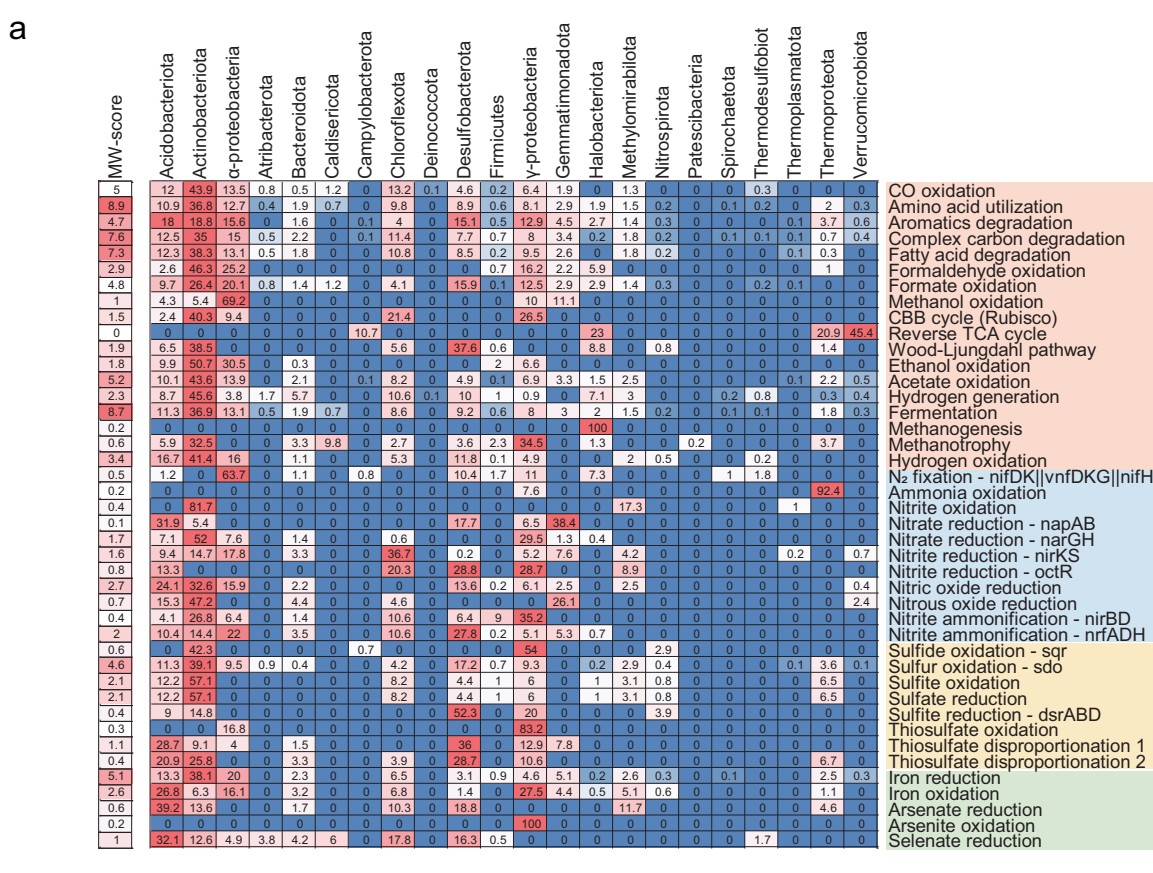

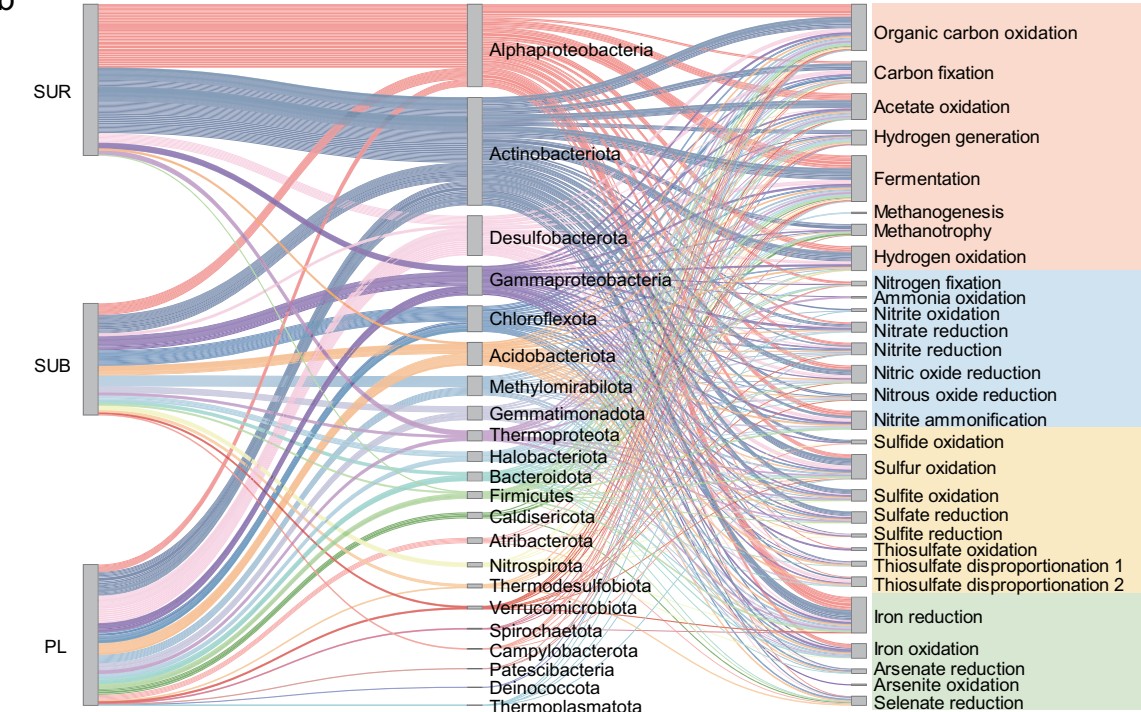

**Fig. 6 | The metabolic weight scores (MW-score) and the percentage contribution of microbial groups to the scores for the genomes on the Tibetan Plateau. a** Heatmap showing the MW-score for each metabolic pathway and the percentage contribution of each microbial groups to the scores. **b** Sankey diagram showing the difference in the contributions of microbial groups to individual biogeochemical processes among three soil layers, with the respective taxonomic classification and category of nutrients. The three columns indicate, from left to right, soil layers, the contribution of each taxonomic group to the metabolic function, and the contribution of each metabolic/biogeochemical cycle, respectively. SUR surface layer, SUB subsurface layer, PL permafrost layer. Source data are provided as a Source Data file.

from deceased cells, may affect the veracity of the conclusions drawn in soil DNA-based investigations remains unresolved[25]. Third, despite the positive associations observed between biogeochemical processes and gene abundance across our study area (Supplementary Fig. 9), there is still a gap between the realistic microbial activity and geochemical processes in situ. In light of these limitations, further studies should seek to employ methods that remove relic DNA or use approaches such as RNA-based metatranscriptomics or protein-based metaproteomics. Furthermore, integrating microbial measurements with biogeochemical data determined in situ could provide a deeper mechanistic understanding of the relationships between microbial and biogeochemical processes in permafrost ecosystems.

In summary, based on systematic measurements of microbial amplicon and metagenomic data obtained from an ~1000 km transect across the Tibetan Plateau—the world's largest permafrost region outside the high latitudes, this study expands our understanding of the large-scale stratigraphic microbial profiles, from the taxonomic, genetic, and genome-centric view, in the understudied alpine permafrost ecosystem (Supplementary Note 4, Supplementary Tables 3, 4). We found that microbial communities were distinct among soil strata, with the alpha diversity decreasing, but β diversity (spatial variability) increasing, with soil depth. Microbial assemblages were predominantly shaped by dispersal limitation and ecological drift, with a heightened emphasis on dispersal limitation within the permafrost layer. These findings illustrate that the ongoing permafrost thawing, particularly the active layer deepening, may cause regional-to global-scale variations in microbial biogeographic patterns and assemblage mechanisms in permafrost ecosystems. We also found that functional genes involved in reduction reactions (such as nitrite reduction, polysulfide reduction, ferric iron reduction, and methanogenesis) were enriched, and microbial taxa involved in redox reactions were more diverse and contributed significantly to community-level metabolic profiles in the deeper layers. These findings highlight the vital role of biogeochemical cycling of ecologically important elements (e.g., nitrogen, sulfur, iron), which serve as electron acceptors for organic matter oxidation and may have profound effects on soil carbon dynamics in permafrost regions. Overall, this study advances our understanding of taxonomic and functional biogeography across soil strata in permafrost regions and underscores the necessity of incorporating vertical variations of microbial attributes in future modeling endeavors to forecast the dynamics of biogeochemical cycles within this critical ecosystem, particularly in the context of climate warming.

## Methods

### Study area and field sampling
Permafrost is distributed extensively across the Tibetan Plateau, covering ~1.1 × 10⁶ km², which constitutes 40% of the total plateau area[61]. Much of the permafrost on the plateau are mainly comprised of discontinuous and sporadic types[62], which were formed during the Last Glaciation Maximum in the late Pleistocene and the Neoglaciation period in the late Holocene[61,63]. The active layer thickness varies across the region, with a mean value of 1.9 m[64]. In 2016, soil samples were collected at 24 sites along a ~1000 km permafrost transect on the plateau[65,66] (Fig. 1a; Supplementary Table 1). At two of these sites, there were problems obtaining a sufficient DNA yield during DNA extraction and so only samples from 22 sites were processed in this study. The mean annual air temperature across all sites ranges from −4.5 to 1.8 °C, and the mean annual precipitation varies from 245 to 504 mm[66]. The primary vegetation types are alpine steppe, alpine meadow, and swamp meadow, which are dominated by *Stipa purpurea* and *Carex moorcroftii*, *Kobresia pygmaea* and *Kobresia humilis*, and *Kobresia tibetica*, respectively[67]. The soil types encompass Cambisols, Calcisols, and Cryosols, as classified by the World Reference Base for Soil Resources[68].

In each site, a 10 m × 10 m plot with five 1 m × 1 m quadrats along the diagonal lines was established[66]. We used a borehole drill to extract soil cores within each quadrat, with core depths varying between 1.5 and 3.5 m according to the active layer thickness. Soil samples in the active layer were collected at depths of 0–10 cm (SUR: surface) and 30–50 cm (SUB: subsurface). Simultaneously, we collected permafrost samples from the uppermost 50 cm thick layer (PL: permafrost layer). During the collection of permafrost soils, we meticulously excluded unfrozen active layer soil and soil from the transitional zone between the active layer and the permafrost deposits[66]. Soil cores were maintained in a frozen state and transported to the laboratory. To prevent potential surface contaminants being introduced during the drilling procedures, the outer layer of each core was scraped with autoclaved knives and chisels[66]. Notably, five replicates within the 10 m × 10 m plot were expected to represent the average condition at each site, which enables regional sampling to assess large-scale patterns of microbial compositional and functional attributes. With this aim, the soil samples from each layer at each site were homogenized through sterile hammering under cold conditions. Afterwards, the composite soils were divided into two parts: one part was subjected to air-drying for subsequent soil physicochemical measurements, the other was preserved at −80 °C for subsequent DNA extraction.

### Characterization of soil physicochemical variables as well as climatic, anthropogenic and plant properties
Soil physicochemical variables were measured to explore the underlying drivers of microbial compositional variations. Specifically, soil moisture was determined by oven-drying 10 g of fresh soil at 105 °C until it reached a constant weight. Soil pH was measured using a pH meter (PB-10, Sartorius, Germany) with a 1:2.5 soil-to-water mixture. Soil texture data were obtained from ref. 65. Soil organic carbon (SOC) content was determined via the potassium dichromate method[69]. A two-step acid hydrolysis procedure was employed to determine soil labile and recalcitrant C pools[70]. Briefly, soil samples underwent initial hydrolysis with 2.5 M $H_2SO_4$ at 105 °C for 30 min. The resulting hydrolysates were separated from the residue, which was then rinsed with distilled water. The supernatant combined with the initial hydrolysate formed labile carbon pool I (mainly polysaccharides). The remaining residue underwent further hydrolysis with 13 M $H_2SO_4$, and was shaken overnight at room temperature. After dilution to 1 M $H_2SO_4$ with distilled water for above hydrolysates, the sample was hydrolyzed at 105 °C for 3 h, constituting labile carbon pool II (mostly cellulose). Finally, the residue was rinsed with distilled water, dried at 60 °C, and identified as the recalcitrant SOC pool. Dissolved organic carbon and total dissolved nitrogen were quantified using a multi-NC-analyzer (Analytik Jena, Thuringia, Germany). The $NH_4^+$-N and $NO_3^-$-N content were determined by a flow injection analyzer (AutoAnalyzer 3 SEAL, Bran and Luebbe, Norderstedt, Germany). Dissolved organic nitrogen (DON) was then calculated by deducting dissolved inorganic N from total dissolved N.

In consideration of the important effects of climatic, plant, and anthropogenic variables on affecting microbial communities, we retrieved climatic variables, consisting MAT and AI, plant variables including plant species richness and NDVI, and human footprint index from public databases. Specifically, MAT was obtained through the spatial interpolation of meteorological information sourced from 73 meteorological stations across the Tibetan Plateau, spanning the period from 1985 to 2014, which matched the sampling time of 2016. An interpolation procedure was conducted within ArcMap 10.6 (Environmental Systems Research Institute, Inc., Redlands, CA, USA), employing the Kriging interpolation technique, and achieving a spatial resolution of 10 km × 10 km. The original MAT data were provided by the China Meteorological Data Sharing Service System (http://cdc.nmic.cn/home.do). The AI was retrieved from the CGIAR-CSI Global-Aridity and Global-PET database (http://www.cgiar-csi.org)[34].

For the plant variables, plant species richness at each site was extracted from the species richness map with 1- km resolution provided by ref. 71. Additionally, maximum-value composite NDVI was used to characterize plant greenness, which were obtained from the Moderate Resolution Imaging Spectroradiometer (MODIS) aboard NASA's Terra satellites (http://neo.sci.gsfc.nasa.gov/) with ~1 km resolution for every 16-day interval between July to August in 2016 (when soil sampling was conducted). To characterize human disturbance of the sampling sites involved in this study, we retrieved the human footprint index from the National Tibetan Plateau Data Center (https://doi.org/10.11922/sciencedb.933)[72]. This index is a quantitative measurement of human pressures on Earth's land surface, which is determined by summing the weighted values (where 0 denotes unpressured and 10 denotes maximum pressure) of five categories of human disturbance (i.e., land use/cover, night-time light, population density, grazing density, and road/railway distributions)[72]. These data were collected at four time points (i.e., 2000, 2005, 2010, and 2015) at 1 km resolution[72], and the scores of human pressure for each grid at the four time points were averaged to represent the human footprint value at our sampling sites. The human footprint values among our sampling sites were lower than those in typical cities on the Tibetan Plateau (Supplementary Fig. 10), suggesting that most of the sampling sites had been subjected to only minor human disturbance.

## DNA extraction, 16S rRNA gene sequencing and analysis

Soil DNA was extracted using the DNeasy PowerMax Soil Kit (Qiagen, Hilden, Germany) according to the manufacturer's protocol, and was subsequently purified by DNeasy PowerClean Pro Cleanup Kit (Qiagen, Hilden, Germany). DNA quality was checked by using 1% agarose and a NanoDrop ND-8000 spectrophotometer (Thermo Fisher Scientific,Waltham, MA, USA). Final DNA concentrations were quantified using PicoGreen (Life Technologies, Grand Island, NY, USA) with a FLUO star OPTIMA fluorescence plate reader (BMG LabTech, Jena, Germany). The primers 515F (5′-GTGCCAGCMGCCGCGGTAA-3′) and 806R (5′-GGACTACHVGGGTWTCTAAT-3′)[73] were used to amplify the hypervariable V4 region of the 16S rRNA gene. Subsequently, the PCR products were assessed through a 2% agarose gel, purified using the AxyPrep DNA Gel Extraction Kit from Axygen Biosciences, and quantified using QuantiFluor™ -ST Fluorometer (Promega, USA). Sequencing libraries were constructed using the TruSeqTM DNA Sample Prep Kit (New England Biolabs, MA, USA) and evaluated by a Qubit@ 2.0 Fluorometer (Thermo Fisher Scientific, MA, USA) and an Agilent Bioanalyzer 2100 system (Agilent Technologies, Waldbron, Germany). Finally, the libraries were sequenced using the Illumina Miseq PE300 platform (Illumina, San Diego, CA, USA).

Paired-end raw sequences were merged using Vsearch v2.15.2[74] with the --fastq_mergepairs function, and quality filtered using the --fastx_filter function with a maximum expected error threshold of 0.01. Filtered reads were then dereplicated using the --derep_fulllength function. Afterwards, unique sequences were denoised and clustered into ASVs using the unoise3 algorithm[75]. The most abundant sequence within each ASV was selected as the representative sequence and aligned against the Silva 138 database for taxonomic annotation (https://www.arb-silva.de/). Mitochondria and chloroplasts were subsequently removed. Finally, we used MUSCLE v5.1[76] to align the representative sequences and constructed the phylogenetic tree with FastTree v2.1.10[77].

## Metagenome sequencing and analysis

DNA extracts were fragmented using a Covaris M220 (Gene Company Limited, China). The resulting fragments had an approximate average size of 400 bp. A paired-end library was prepared using NEXTFLEX Rapid DNA-Seq (Bioo Scientific, Austin, TX, USA) and sequenced on an Illumina NovaSeq 6000 (Illumina Inc., San Diego, CA, USA) at Majorbio

Bio-Pharm Technology Co., Ltd. (Shanghai, China) using the NovaSeq 6000 S4 Reagent Kit v1.5 (300 cycles) according to the manufacturer's instructions. The workflow for the analysis of metagenomic sequence data is shown in Supplementary Fig. 11. Specifically, fastp v0.21.0[78] was employed to perform quality control and adapter trimming. After the quality control process, we obtained 995.8 gigabases (Gb) of metagenomic clean reads, with average value of 14.5 Gb data per sample (range from 12.3 to 17.5 Gb). Filtered reads were then assembled to contigs using megahit v1.2.9[79] with default k-mers. Only contigs with a length longer than 500 bp were translated to protein-coding ORFs using Prodigal v2.6.3[80] with metagenomic mode as default. Then, CD-HIT v4.8.1[81] was used to remove redundancy and generate a gene catalog by clustering ORFs at sample level and globally with a 95% sequence identity cutoff. The abundance of each gene (gene counts per KEGG Orthology-KO) was quantified using Salmon v1.5.1[82] and normalized to transcript per million (TPM) based on the gene length and sequencing depth. Then genes were mapped to the eggNOG 5.0 database for functional annotation using eggnog-mapper v2.1.3[83] with the DIAMOND mode.

Assembled contigs were further binned to metagenome-assembled genomes using metaWRAP v1.3.2[84] with the self-implemented MaxBin2[85], metaBAT, and metaBAT2[86] binning modules. MAGs were refined with the bin_refinement module in metaWRAP and deduplicated by dRep v3.4.3[87]. During the bin deduplication process, the quality of MAGs was evaluated by checkM v1.2.2[88], and good quality MAGs (≥70% completeness and ≤10% contamination) were retained for further analysis. The bin abundance was quantified by Salmon, and their taxonomic information was annotated using GTDB-Tk v2.1.1 with the GTDB r207 database[89]. The recruitment rates of the MAGs were determined by aligning the qualified reads to all MAGs using bowtie2 v2.3.5.1[90]. To determine the metabolic potential of the MAGs, we first used the Prodigal module of METABOLIC v4.0[48] to predict the protein-coding ORFs on all genomes. The hmmsearch program was then performed to annotate proteins against the HMM databases (KEGG KOfam, Pfam, TIGRfam, and custom HMMs) implemented within METABOLIC v4.0 with the default options. To explore the functional capacity of each metabolic pathway, metabolic weight scores (MW-scores) were calculated based on the results of metabolic profiling and gene coverage obtained from metagenomic read mapping (Supplementary Note 5). Moreover, to reveal the variations of metabolic groups and their corresponding contributions to the metabolic pathways across soil depth, we first resampled (100 times) MAGs in subsurface and permafrost layers to the same MAGs number as in the surface layer to minimize the potential bias induced by different MAGs sizes. We then computed the percentage contribution of each microbial phylum to the MW-score of each metabolic pathway with each resampled dataset (Supplementary Note 5). The average value was used as the final value of the percentage contribution. High percentage contribution indicates that microbial groups can better represent the function from both gene presence and abundance[48].

## Statistical analyses

A series of statistical analyses were conducted to investigate the microbial stratigraphic profile. Specifically, we used the estimate_richness function within the phyloseq[91] package to calculate the Shannon-Wiener index. Faith index was determined by the pd function in the picante[92] package to characterize microbial phylogenetic diversity. To explore the microbial spatial variation (β diversity), we determined the Bray-Curtis metric to characterize taxonomic variation. The βMNTD was measured by the comdistnt function of the picante[92] package to characterize the phylogenetic variation. Pairwise Wilcoxon test was employed to compare the difference in above metrics among the soil layers by using the compare_means function of the ggpubr[93] package. Three multivariate statistical analyses, including permutational multivariate analysis of variance (Adonis), analysis of

similarities (ANOSIM), and multi-response permutation procedure (MRPP), were conducted to test the overall effect of soil layers on microbial community structure[94]. Furthermore, the specificity and occupancy of each ASV were calculated in each soil layer to characterize specialist ASVs[29,95]. Specificity is operationally defined as the average abundance of a given ASV within a set of habitat samples, while occupancy is characterized as the relative frequency with which ASV occurs within the same set of habitat samples[29,95]. ASVs with specificity and occupancy greater or equal to 0.7 were defined as specialist species, which indicated that they were specific to a habitat and common in most sites[95].

To explore the underlying drivers of the microbial community structure, we first employed variable clustering to assess the collinearity of the eighteen environmental variables and remove redundant variables (variables with a high correlation (Spearman's $\rho^2 > 0.7$)) as calculated by the varclus procedure in the Hmisc[96] R package. Thirteen non-redundant variables (e.g., AI, pH, NDVI; Supplementary Fig. 2) were then employed to detect correlations between the microbial community structure and environmental variables using the partial Mantel test. In this analysis, microbial compositional variation among soil samples was determined using Bray-Curtis distance, while the environmental dissimilarity matrices were calculated using Euclidean distance. Both dissimilarity matrices were computed using the vegdist function of vegan[97] package. Moreover, a general framework of phylogenetic-bin-based null model analysis (iCAMP)[98] was used to quantify the effects of ecological processes in shaping the microbial community. The framework employed the β-net relatedness index and the Raup-Crick index to infer phylogenetic beta diversity and taxonomic beta diversity[98,99]. The relative importance of five assembly processes: heterogeneous selection, homogeneous selection, dispersal limitation, homogenizing dispersal, and drift (and others such as diversification, weak selection, and/or weak dispersal) was calculated, and their differences among soil layers determined by means of the Wilcoxon test.

To better elucidate the difference in functional profiles across three soil layers, we initially identified genes associated with key biogeochemical processes related to essential elements such as carbon, nitrogen, sulfur, and iron (see Supplementary Data 2). We then employed a generalized linear model (GLM) with a negative binomial distribution to estimate differences in gene expression among these soil layers. Significance was determined using a Benjamini-Hochberg false discovery rate with $P$ value threshold of <0.05. The GLM modeling was carried out using the glmFit function from the edgeR[100] package. Further, we utilized principal coordinate analysis to assess variations in genome composition across the soil layers. Multivariate statistical analyses were conducted to evaluate the overall effect of soil layers on genome composition. Additionally, we employed the Wilcoxon test to detect differences in mean genome size amongst the three soil layers. All $P$ values generated by Wilcoxon test in this study were adjusted using the Benjamini-Hochberg false discovery rate procedure. All the statistical analyses were performed using R 4.0.3[101].

### Reporting summary
Further information on research design is available in the Nature Portfolio Reporting Summary linked to this article.

## Data availability
The 16S rRNA gene sequence data and the metagenomic sequence data generated in this study are deposited in the NCBI Sequence Read Archive (SRA) database under the BioProject number PRJNA1037019. Data for the main results of this study are publicly available from https://doi.org/10.5281/zenodo.11829921. Source data are provided with this paper.

## Code availability
The R code used for the statistical analyses is available at https://github.com/kangluyao/Microbes_in_Tibetan_permafrost and https://doi.org/10.5281/zenodo.11829921[102].

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

## Acknowledgements

We thank the members of the IBCAS Sampling Team (Drs. Dan Kou, Yongliang Chen, and Chao Mao) for permafrost sampling on the Tibetan Plateau. We also thank Dr. Changjin Cheng (South China Botanical Garden, the Chinese of Academy of Sciences) for providing data on plant species richness. This work was supported by the National Key Research and Development Program of China (2022YFF0801901), the National Natural Science Foundation of China (31988102 and

32425004), and New Cornerstone Science Foundation through the XPLORER PRIZE.

## Author contributions

Y.Y. and L.K. conceived the study. L.K. and Y.S. analyzed the data. L.K. performed the laboratory experiments. L.K. and Y.Y. wrote the manuscript. R.M., D.Z., S.Q., L.C., L.W., and Y.P. contributed to subsequent revisions.

## Competing interests

The authors declare no competing interests.
