## [Peer Review File · Nature Communications]

Metagenomic insights into microbial community structure and metabolism in alpine permafrost on the Tibetan PlateauREVIEWER COMMENTS

Reviewer #1 (Remarks to the Author):

The authors have conducted an analysis of the microbial community structure and metabolic potential in permafrost soils of the Tibetan Plateau, utilizing 16S rRNA and shotgun metagenomic sequencing. Numerous previous studies have explored similar aspects of microbial communities and functional potentials in permafrost soils, yielding comparable findings. Consequently, this work appears to be somewhat redundant and overly descriptive. While the presentation of findings is commendable, the manuscript lacks sufficient originality, warranting consideration for rejection.

The critique extends to the perception that the work is predominantly descriptive, with limited emphasis on establishing a meaningful linkage between microbial and geochemical data. A more profound mechanistic understanding of the relationships between microbial processes and biogeochemical processes in the permafrost soil ecosystem would greatly enhance the scientific contribution.

Reviewer #2 (Remarks to the Author):

This is a very interesting article with a large and useful data set and thoughtful analysis. The community assembly framework is a nice approach, but this framework could be made clearer and more self-consistent. My specific comments follow:

Abstract (L47) says higher redox potential in permafrost – this seems contradictory, should logically be more negative, consistent with the presence of genes for alternative electron accepting processes.

The community assembly framework is interesting, but only drift and dispersal are mentioned in the introduction, while selection (and speciation) are not mentioned until later. Selection is invoked as an explanation for lower alpha diversity in permafrost, but isn't included in the framework developed in the introduction.

The hypotheses sound suspiciously like a post hoc statement of results. The development of the hypothesis 1 in the intro makes sense but could use a little more explanation (presumably the argument that freeze-thaw events lead to drift is that fluctuating populations experience bottlenecks, though this should also actively select for resistant species). Hypothesis 2 seems after-the-fact because one would expect the lower parts of the active layer (30-50 cm) to also be anoxic and feature the same alternative e- accepting processes predicted for permafrost (and fermentation, too). Given that metabolism is higher in the active layer, wouldn't one predict more reliance on alternative e- acceptors compared to permafrost, where slower metabolic rates would lessen their scarcity? In fact, it seems the metagenomic results gloss over the subsurface layer and focus solely on the surface vs permafrost layers (more on this below). I am not suggesting that the hypotheses should be changed, but if these are truly a priori hypotheses, the theory behind them should be made as clear as possible. Also, the use of the phrase, "could be" isn't appropriate for hypotheses, as they are predictions that will be tested, not just possibilities to be explored.

L181, aridity index should be defined explicitly if it is an important predictor.

L136-139, prolonged evolution in permafrost diversifies community? This seems to contradict the lower alpha diversity. This is an interesting idea, though, and could maybe be tested by focusing on specific taxonomic groups and comparing permafrost sequences to active layer sequences in the same clade. Within a family of bacteria, do permafrost species create distinct subclades?

The observation that AI predicts community structure in permafrost provides evidence that selection by environmental factors is also important, in addition to drift and dispersal. The

community assembly model results are interesting, but it should also be noted that not all of the potentially important explanatory environmental factors were included. See comments below about redox, disturbance and plant community.

L204-205, if any of the sites are subject to recent human disturbance, it would be good to document this and include it in the analysis. Likewise, was vegetation included in the analysis?

L302-307, It is true that Acidobacteria are slow-growing and stress-resistant, but they are plentiful in soils from all over the world, not just the Arctic and Antarctic. I suspect that many Chloroflexi in the permafrost are associated with fermentation. Either way, it would be useful to include a discussion of fermentative processes, as these co-dominate anoxic soil layers, along with the alternative electron accepting processes discussed here.

Figure 5b shows that permafrost and subsurface are metabolically more similar to each other than to the surface. This is evidence that redox conditions deeper in the soil drive selection. See for example Environmental Microbiology Reports 7.4 (2015): 649-657. As redox was not included in the 10 explanatory environmental variables, this should be discussed.

Finally, check the grammar - there are some awkward phrases throughout.

Reviewer #3 (Remarks to the Author):

This is an impressive and important study describing the microbial communities across a large section of permafrost associated soils in the Tibetan Plateau. The study includes sampling of both the active layer as well as associated permafrost layer providing a link between the frozen environment and the environment. The results of this study are noteworthy in the scope of the data set providing such a large sampling transect in the north, an area being warmed by climate more rapidly than other places on the planet. Studies like this provide useful information on the microbial community and in this case the functional potential of that community.

The authors propose two logical hypotheses to frame the study of this system. They find broad support for both. The authors find support that drift likely imposed by dispersal limitations is driving the alpha and beta diversity gradients across the soil depth. While this may not be particularly novel, the geographic area sampled and the breadth of the study provides the novelty by increasing the sampling and knowledge of this understudied area. Using the functional potential based on the metagenomic sequencing, they find a potential enriched role for biogeochemical cycling found in permafrost soils. This result provides useful data for a better understanding of the impacts of climate change on global carbon dynamics as permafrost represents a large source of carbon.

I agree with the authors that three ecological processes are likely driving the community structure. However, you state that the order of importance is different for the permafrost layer compared to the surface layer. In your methods you indicate that you tested for differences among soil layers (L537) and Figure 3C looks as though you compared soil layers within each ecological mechanism. Did you test for differences in mechanism within the layer as that appears to be the result you are reporting? Based on the figure, it looks like the contributions of dispersal limitation and drift are equally important.

Figure 3C uses the term "Drift and Others" but this doesn't exactly align with the text where you use "ecological drift". Using the same terminology in both would help the reader understand better.

When discussing the MAGs found across the different soil layers, you found that average genome size decreased with increasing layer depth. While your explanation about compact/simplified genomes is consistent with this evidence, are there other explanations? I wonder what the recovery rate of MAGs was across the three layers that were sampled? Are there any potential biases in the number or quality of MAGs between the layers? Providing more details on the

different layers may help support your explanation but may also provide alternatives for your observations.

While I understand that the plot provided in Figure 5A is somewhat common, the information could be presented differently to enhance the reader's understanding. The phylogenetic tree is much too small to be of any use. The other information about metabolic genes is also too condensed to be helpful.

The methods section is well written and adequately describes the extensive bioinformatic pipelines. I appreciate the authors' detailed descriptions that facilitate reproducibility. Thank you for making your code available on GitHub.

The introduction states (L101) that you set up 22 sites along your permafrost transect, but in the methods (L372) you state that 24 sites were sampled. Based on the data tables, it appears that the introduction is the correct value.

Can you provide a rationale for homogenizing the soil samples from the five quadrats within a site? It seems logical for the metagenomic data, but you missed out on measuring within site variation for the 16S amplicon portion of the study. Providing this information may help others understand the design better.

Please consider also citing #95 at L514 for a clearer explanation of the methods for calculating the specificity and occupancy. In addition, can you clarify if you are using ASVs as the unit here or are you collapsing ASVs using the assigned taxonomy into species? The text appears to use species and ASV interchangeably when they are not.

The SRA data doesn't appear to be available yet. Please ensure this is publicly available upon publication.

Minor comment

L123 change ascending to increasing to avoid a misunderstanding

Responses to Reviewer #1

[Comment 1] The authors have conducted an analysis of the microbial community structure and metabolic potential in permafrost soils of the Tibetan Plateau, utilizing 16S rRNA and shotgun metagenomic sequencing. Numerous previous studies have explored similar aspects of microbial communities and functional potentials in permafrost soils, yielding comparable findings. Consequently, this work appears to be somewhat redundant and overly descriptive. While the presentation of findings is commendable, the manuscript lacks sufficient originality, warranting consideration for rejection.

[Response] Thanks for the reviewer's comments. To demonstrate the novelty of this manuscript, we summarized current studies related to microbes in permafrost regions in term of functional potentials and community assembly mechanisms. **In the aspect of microbial functional potentials**, current studies can be broadly categorized into three groups based on the research questions and methods. **The first major category explored the response of permafrost microbial functional potentials to experimental warming.** For instance, Mackelprang *et al.* (2011) had revealed that multiple genes involved in cycling of C and nitrogen shift rapidly during permafrost thawing by incubation. Xue *et al.* (2016), Johnston *et al.* (2019) and Wu *et al.* (2022) explored the microbial response to ecosystem warming experiment *in situ* at CiPEHR, Eight Mile Lake region, Interior Alaska. **Another major category focused on variations of microbial attributes along permafrost thawing sequence.** For example, Mondav *et al.* (2014), and McCalley *et al.* (2014) revealed the characteristics of methanogenesis, while Singleton *et al.* (2018) explored the methanotrophs along the permafrost thaw sequence. Most of these studies are from the permafrost thaw sequence at Stordalen Mire, Abisko, Sweden. **The other studies are grouped due to their special focus** such as the effects of wildfires on tundra microbes (Taş *et al.*, 2014), microbial survival strategies in ancient permafrost (Mackelprang *et al.*, 2017). All these studies differ in terms of their concerns, methodologies, and study regions (Table R1). **In the aspect of community assembly mechanisms**, current studies mostly

concerned the variations in assemblage mechanisms during permafrost thawing process (Doherty *et al.*, 2020; Feng *et al.*, 2020; Mondav *et al.*, 2017) or solely focus on the contribution of ecological processes in intact permafrost deposits (Bottos *et al.*, 2018; Hu *et al.*, 2015) (Table R2). The underlying ecological processes of permafrost microorganisms is still in infancy (Ernakovich *et al.*, 2022). In this study, we focused more on revealing the variations of biogeographic patterns, the underlying assemblage mechanisms, and metabolic potentials from the surface active layer to deep permafrost layer over the large scale.

Despite previous studies have explored microbial communities and functional potentials in permafrost soils, this study is innovative in three aspects. First, preceding investigations concerning permafrost microorganisms have been predominantly constrained to site-specific level (Table R1), our study provided the first large-scale stratigraphic characteristics of microorganisms in permafrost regions. To our knowledge, only two studies, conducted by Waldrop *et al.* (2023) and Vishnivetskaya *et al.* (2020), have revealed gene variation and photosynthetic microorganisms, respectively, across the pan-arctic permafrost regions based on the synthesis of sequence data. The data set involved in our study was derived from systematic measurements among three soil layers along a ~1,000 km transect, which can provide insightful information on the microbial composition and functional potential in permafrost region. Consistent with our point, the other two reviewers also recognized the novelty of our manuscript as follows: “reviewer#2 [Comments #1] *This is a very interesting article with a large and useful data set and thoughtful analysis*” and “reviewer#3 [Comments #1] *This is an impressive and important study describing the microbial communities across a large section of permafrost associated soils in the Tibetan Plateau.... The results of this study are noteworthy in the scope of the data set providing such a large sampling transect in the north, an area being warmed by climate more rapidly than other places on the planet. Studies like this provide useful information on the microbial community and in this case the functional potential of that*

community”.

Second, most current studies concerning microorganisms were mainly confined to high-latitude permafrost region, with little evidence from high-altitude permafrost region (Table R1). This study deciphered the microbial attributes in Tibetan alpine permafrost region, which provide insightful view into microbial communities and functional potentials in the largest permafrost region at the mid- and low latitudes of the world. This novelty was also recognized by reviewer#3, “[Comments #2] the geographic area sampled and the breadth of the study provides the novelty by increasing the sampling and knowledge of this understudied area”.

Third, based on the systematic measurements, combining with the thoughtful analysis (e.g., null model analysis, occupancy and specificity analysis, and metabolic weight scores), our study provided several new findings on permafrost microbes. 1) We found the lesser effects of environmental variables in permafrost layer comparing to active layer, suggesting the weaker response of microbes to the environment selection in permafrost soils, which may be ascribed to their survival strategy such as dormancy. 2) We found that genes participating in reduction reactions (e.g., dissimilatory nitrate reduction, denitrification, ferric iron reduction, sulfide reduction, tetrathionate reduction) were enriched in permafrost soils, implying that microbes colonizing in permafrost soils possessed specific metabolic capabilities in spite of the harsh conditions in frozen soil. These findings advanced our understanding of microbial profiles in permafrost regions.

Collectively, this manuscript expands our understanding of large-scale stratigraphic microbial profiles from taxonomic, genetic, and genome-centric view based on the systematic measurements along a ~1,000 km transect at the largest alpine permafrost region (i.e, Tibetan Plateau) in the mid- and low latitudes of the world. We have added the novelty statement in our revised MS (Page 19, lines 417-

422. Supplementary note 4, Supplementary Tables 3-4). Thanks for your understanding!

Table R1: Summary of current studies involved in permafrost microorganisms based on meta-omics data. MG: metagenomics. MT: metatranscriptomics. MP: metaproteomics. The first major category explored the response of permafrost microbial functional potentials to experimental warming. Second major category focused on the shift patterns of microbial attributes along permafrost thawing sequence. The other studies are grouped due to their special focus (category 3). “Measured” in data source column means that data are produced with the same procedures, while “Synthesis” means that data are collected from different studies/projects.

References	Region	No. of	Category	Data source	Methods
Mackelprang et al. (2011) Nature	Pan-Arctic	1	1	Measured	Amplicon, MG, qPCR
Tveit et al. (2015) Proc. Natl. Acad. Sci. U. S. A.	Pan-Arctic	1	1	Measured	MT, MG
Xue et al. (2016) Nat. Clim. Chang.	Pan-Arctic	1	1	Measured	Amplicon, GeoChip, MG
Johnston et al. (2019) Proc. Natl. Acad. Sci. U. S. A.	Pan-Arctic	1	1	Measured	Amplicon, MG
Wu et al. (2022) Mol. Ecol.	Pan-Arctic	1	1	Measured	Amplicon, GeoChip, MG
Mondav et al. (2014) Nat. Commun.	Pan-Arctic	1	2	Measured	Amplicon, MG
McCalley et al. (2014) Nature	Pan-Arctic	1	2	Measured	Amplicon, MG
Hultman et al. (2015) Nature	Pan-Arctic	1	2	Measured	Amplicon, MG, MT, MP
Singleton et al. (2018) ISME J.	Pan-Arctic	1	2	Measured	MG, MT
Woodcroft et al. (2018) Nature	Pan-Arctic	1	2	Measured	Amplicon, MG, MT
Yergeau et al. (2010) ISME J.	Pan-Arctic	1	3	Measured	16S, qPCR
Tveit et al. (2013) ISME J.	Pan-Arctic	2	3	Measured	MG, MT
Taş et al. (2014) ISME J.	Pan-Arctic	1	3	Measured	Amplicon, MG
Geisen et al. (2015) ISME J.	Pan-Arctic	1	3	Measured	MT
Mackelprang et al. (2017) ISME J.	Pan-Arctic	1	3	Measured	Amplicon, MG
Müller et al. (2018) Environ. Microbiol.	Pan-Arctic	1	3	Measured	Amplicon, MG
Taş et al. (2018) Nat. Commun.	Pan-Arctic	1	3	Measured	Amplicon, MG
Wu et al. (2021) Environ. Sci. Technol.	Pan-Arctic	1	3	Measured	Amplicon, MG
Wu et al. (2023) Environ. Microbiome	Pan-Arctic	2	3	Measured	Amplicon, MG
Waldrop et al. (2023) ISME J.	Pan-Arctic	18	3	Synthesis	MG
Vishnivetskaya et al. (2020), FEMS Microbiol. Ecol.	Pan-Arctic	11	3	Synthesis	Amplicon, MG
Tang et al. (2023) FEMS Microbiol. Ecol.	Tibetan Plateau	4	3	Measured	Amplicon, qPCR
This study	Tibetan Plateau	22		Measured	Amplicon, MG

Table R2: Summary of current studies related to microbial assemblage mechanisms in permafrost soils.

References	Region	No. of sites	Information of site
Hu et al. (2015) PLoS One	Tibetan Plateau	1	Intact permafrost
Bottos et al. (2018) FEMS Microbiol. Ecol.	Pan-Arctic	1	Intact permafrost
Doherty et al. (2020) Front. Microbiol.	Pan-Arctic	1	Thawed permafrost
Mondav et al. (2017) Environ. Microbiol.	Pan-Arctic	1	The active layer of thawing permafrost landscapes
Feng et al. (2020) Microbiome	Pan-Arctic	1	The active layer of thawing permafrost landscapes
Wu et al. (2022b) Sci. Total Environ.	Tibetan Plateau	1	The active layer of thawing permafrost landscapes
Wu et al. (2022a) Mol. Ecol.	Pan-Arctic	1	The active layer of thawing permafrost landscapes
This study	Tibetan Plateau	22	The active layer and intact permafrost

[Comment 2] The critique extends to the perception that the work is predominantly descriptive, with limited emphasis on establishing a meaningful linkage between microbial and geochemical data. A more profound mechanistic understanding of the relationships between microbial processes and biogeochemical processes in the permafrost soil ecosystem would greatly enhance the scientific contribution.

[Response] Thanks for the reviewer's comments! We agree with the reviewer's point that establishing the linkages between microbial and geochemical data can provide a more profound mechanistic understanding of the relationships between microbial processes and biogeochemical processes. To link the microbial and biogeochemical processes, we explored the relationships between the microbial data obtained in this study and the biogeochemical processes determined in previous studies from our research group (*i.e.*, the cumulative CO₂ release, the methane production potential, and the nitrogen cycling) (Mao *et al.*, 2020; Qin *et al.*, 2021; Song *et al.*, 2021). We found that the genes involved in hemicellulose (*i.e.*, arabinofuranosidase, beta-mannanase, xylanase, xylose isomerase), cellulose (*i.e.*, beta-glucosidase), chitin (*i.e.*, acetylglucosaminidase, endochitinase, exochitinase), and pectin (*i.e.*, pectinase) degradation showed significant positive relationships with the cumulative CO₂ release (Fig. R1). By contrast, the genes encoding methyl-coenzyme M reductase, particulate methane monooxygenase, and soluble methane monooxygenase displayed no significant associations with methane production (Fig. R1). In regard to the nitrogen cycling, we found that the genes *gdh_k00261* and *gdh_k00262* were positively associated with the gross nitrogen mineralization rate, and the *hao* gene was significantly related to nitrification (Fig. R1). **We have added these results in our revised MS** (Page 19, lines 406-415; Supplementary Fig. 9).

Although we observed significant associations between microbial and geochemical data, the biogeochemical processes determined via incubation experiments may differ from those measured *in situ*, and it would be a great challenge to measure the biogeochemical processes (e.g., fluxes of CO₂ and CH₄) along a ~1,000 km transect in

our study, particularly for the deep permafrost deposit. Due to this point, we have added a statement regarding the limitations on the linkage between microbial and geochemical data in our study as follows: *“Third, despite the positive associations observed between biogeochemical processes and gene abundance across our study area (Supplementary Fig. 9), there is still a gap between the microbial and geochemical measurements, particularly regarding the linkage between realistic microbial activity and biogeochemical processes in situ. In light of these limitations, further studies should seek to employ methods that remove relic DNA or use approaches such as RNA-based metatranscriptomics or protein-based metaproteomics. Furthermore, integrating microbial measurements with biogeochemical data determined in situ could provide a deeper mechanistic understanding of the relationships between microbial and biogeochemical processes in permafrost ecosystems”* (Page 19, lines 406-415). Thanks for your understanding!

Fig. R1: The relationships between gene relative abundance and biogeochemical processes. **a-i** Linear regression relationships between cumulative CO₂ release and the relative abundance of genes encoding carbon degradation. **j-l** The relationships between the CH₄ production and the relative abundance of genes involved in methane production and oxidation. **m-o** The relationships between the nitrogen mineralization and nitrification rate and the relative abundance of genes participated in nitrogen cycling. Cumulative CO₂ release in permafrost soils was measured via 400-day incubation at 5°C by Qin *et al.* (2021) CH₄ production potentials of subsurface and permafrost soils was measured through 24 h incubation at 4°C using the subsurface and permafrost soil samples by Song *et al.* (2021). Nitrogen cycling processes of three soil layers were measured via 24 h incubation at 5°C using ¹⁵N labeling by Mao *et al.* (2020) Shadow area shows 95% confidence interval. All these biogeochemical processes are measured using the same soil samples as in this study. Shadow area shows 95% confidence interval.

Responses to Reviewer #2

[Comment 1] *This is a very interesting article with a large and useful data set and thoughtful analysis. The community assembly framework is a nice approach, but this framework could be made clearer and more self-consistent.*

[Response] Thanks for the reviewer's positive comments. These comments listed below guided us to conduct more rigorous revision on the manuscript. We really appreciate this professional review which greatly improved our paper. Detailed modifications about the community assembly framework and others, please see our responses to the following comments.

[Comment 2] *Abstract (L47) says higher redox potential in permafrost – this seems contradictory, should logically be more negative, consistent with the presence of genes for alternative electron accepting processes.*

[Response] Sorry for the confusion. The redox potential in permafrost is indeed lower. To avoid the confusion, we have now revised the sentence as follows: “*Microbial groups involving the alternative electron accepting processes are more diverse and contribute highly to community-level metabolic profiles in the subsurface and permafrost layers, reflecting **the lower redox potential** and more complicated trophic strategies for microorganisms in deeper soils*” (Page 2, lines 44-48)

[Comment 3] *The community assembly framework is interesting, but only drift and dispersal are mentioned in the introduction, while selection (and speciation) are not mentioned until later. Selection is invoked as an explanation for lower alpha diversity in permafrost, but isn't included in the framework developed in the introduction.*

[Response] Following the reviewer's comments, and combining with the [Comment #4], we have incorporated the selection process into the framework in the *Introduction* section as follows: “*Specifically, recurrent freeze-thaw cycles in the active layer act as disruptive events.... **Additionally, soil structure, moisture, and substrate availability vary with the freeze-thaw cycles^{10,11}, potentially leading to niche differentiation***”

(selection) due to the difference in microbial physiologic traits¹¹. The frozen permafrost soils.... Meanwhile, niche selection mediated by harsh conditions (e.g., oxygen depletion, nutrient scarcity, or freezing temperature) may reduce microbial diversity in permafrost soils during prolonged frozen periods, while speciation via mutation may increase it^{1,14}. However, the strength of both niche selection and speciation may be muted in the permafrost layer by lower metabolic activity or a resistant survival strategy, such as dormancy¹⁵” (Page 4, line 76-90).

[Comment 4] The hypotheses sound suspiciously like a post hoc statement of results. The development of the hypothesis 1 in the intro makes sense but could use a little more explanation (presumably the argument that freeze-thaw events lead to drift is that fluctuating populations experience bottlenecks, though this should also actively select for resistant species).

[Response] We have added more explanations to our hypothesis 1, especially for the selection process in the community assembly framework. Given that the freeze-thaw events occur in the surface layer (Fig. R2), we have added more explanations about the drift and selection process mediated by the freeze-thaw events in surface layer as follows: “Specifically, recurrent freeze-thaw cycles in the active layer act as disruptive events that directly induce fluctuations in microbial population sizes via demographic processes^{8,9}, which may consequently lead to prominent effects of ecological drift on the microbial assemblages. Additionally, soil structure, moisture, and substrate availability vary with the freeze-thaw cycles^{10,11}, potentially leading to niche differentiation (selection) due to the difference in microbial physiologic traits¹¹” (Page 4, lines 76-82)

Fig. R2: The number of diurnal freezing-thawing events within one year along the soil depth on Tibetan Plateau. The diurnal freezing-thawing events are recorded when the daily minimum soil temperature drops below 0°C, and the daily maximum soil temperature reaches 0°C or higher (Baker & Ruschy, 1995). Soil temperature data were derived from seventeen sites on the Tibetan Plateau, which recorded by Yang *et al.* (2019), Wei *et al.* (2021), and our research group.

[Comment 5] Hypothesis 2 seems after-the-fact because one would expect the lower parts of the active layer (30-50 cm) to also be anoxic and feature the same alternative e- accepting processes predicted for permafrost (and fermentation, too). Given that metabolism is higher in the active layer, wouldn't one predict more reliance on alternative e- acceptors compared to permafrost, where slower metabolic rates would lessen their scarcity? In fact, it seems the metagenomic results gloss over the subsurface layer and focus solely on the surface vs permafrost layers (more on this below). I am not suggesting that the hypotheses should be changed, but if these are truly a priori hypotheses, the theory behind them should be made as clear as possible. Also, the use of the phrase, "could be" isn't appropriate for hypotheses, as they are predictions that

will be tested, not just possibilities to be explored.

[Response] Very insightful comments! **Following the reviewer's comments, we have changed the phrase "could be" and also used the phrase "with increasing soil depth" to include the subsoils information in our hypothesis 2 as follows:** "*With these measurements, we aimed to test.... 2) The relative abundance of genes participating in reduction reactions related to alternative electron acceptors is higher, and the taxa engaged in the redox reactions become more diverse **with increasing soil depth**" (Page 6, lines 124-127). **Additionally, we have added more explanation to our hypothesis 2 as follows:** "*One notable characteristic of deep permafrost soils is that the microbial community structures are affected by the redox status due to the limited oxygen availability^{16,17}. In such conditions, microorganisms are more likely to engage in anaerobic respiration and fermentation for soil organic matter degradation¹⁸. These processes necessitate a series of reductive reactions involving diverse alternative electron acceptors, such as nitrate, sulfate, ferric iron, carbon dioxide, and small organic molecules². Therefore, genes associated with reduction reactions pertinent to the aforementioned elements may become more prevalent, and the corresponding taxa involved in these reduction pathways may be more diverse, showing a greater contribution to the community-level metabolic profiles with soil depth. Alternatively, microorganisms may also be less reliant on alternative electron acceptors due to the slower microbial activities in deeper soils¹⁹. Despite the recognitions of these microbial processes, there exists a dearth of knowledge concerning their variability across different soil strata, and that of their corresponding functional groups, as well as their respective contributions to the overall community metabolism across permafrost regions"* (Pages 4-5, lines 95-110).*

Following the reviewer's comments, we have also added more results and discussions about the microbial metabolic traits in subsoils at functional gene level as follows: "*In contrast to the surface layer, we observed an elevation in the abundance of genes linked to the degradation of hemicellulose, cellulose, and pectin in the*

subsurface and permafrost layers (Fig. 4a)...Given that both the subsurface and permafrost layer are more anoxic than the surface layer, microorganisms colonized in these layers tend to utilize the alternative electron acceptors (e.g., NO_3^- , Fe^{3+} , SO_4^{2-}) to facilitate the anaerobic degradation process^{2,18}” (Pages 13-14, lines 274-294). We have also added more results and discussions about the microbial metabolic traits in subsoils at genome levels as follows: “In contrast to the surface layer observations, we found that not only Actinobacteria but also more diverse taxa such as γ -proteobacteria, Chloroflexota, Methylophilota, and Acidobacteriota occupied a significant functional fraction related to the redox reactions in the subsoils and permafrost soils (Fig. 6b; Supplementary Note 3)...These various taxa, which have adapted to thrive in specific redox niches, ultimately co-dominant the biogeochemical processes in deeper soils¹⁷. Collectively, these results highlighted the diverse microbial species engaged in redox reactions and the more complicated trophic strategies for microorganisms in subsoils and permafrost deposits.” (Pages 16-18, lines 355-389).

[Comment 6] L181, aridity index should be defined explicitly if it is an important predictor.

[Response] Following the reviewer’s comment, we have added an explanation of the aridity index as follows: “Most of the selected environmental variables exhibited significant correlations with microbial compositional variations, with pH and the aridity index (**AI: determined by dividing mean annual precipitation by mean annual potential evapotranspiration³²**) emerging as the primary predictors of microbial assemblages (Fig. 3a)” (Page 9, lines 200-205).

[Comment 7] L136-139, prolonged evolution in permafrost diversifies community? This seems to contradict the lower alpha diversity. This is an interesting idea, though, and could maybe be tested by focusing on specific taxonomic groups and comparing permafrost sequences to active layer sequences in the same clade. Within a family of bacteria, do permafrost species create distinct subclades?

[Response] Sorry for the misuse of the word “diversifies”. Initially, we intended to express that microbial communities in permafrost deposit diverge from one another (increasing taxonomic/phylogenetic beta diversity, as shown in Fig. 1d, e). To eliminate the confusion, we have revised the sentence as follows: “*Over their prolonged evolution within the permafrost, microbial communities are highly isolated, particularly as a result of the soil freezing which acts as a physical barrier²⁴. Consequently, microbial communities at different sites will **diverge** from one another, resulting in significant spatial variations*” (Page 7, lines 157-160).

Nevertheless, following the reviewer’s suggestions, we have chosen five families with highest relative abundance to test the reviewer’s concern that whether permafrost species create distinct subclades. Specifically, we used mean pairwise distance (MPD) (Webb, 2000) metric to determine the phylogenetic distance between taxa within the family for each sample, then conducted pairwise Wilcoxon test to compare the difference in phylogenetic distance among the three layers. Our results showed that there was no general pattern about the phylogenetic distance with increasing soil depth (Fig. R3). Some taxa such as KD4-96 and Nitrosomonadaceae even displayed a decreasing trend in phylogenetic distance with the increasing soil depth (Fig. R3). These additional analyses suggest that we may not be able to conclude that distinct subclades are produced in permafrost deposits.

Fig. R3: Comparison of the mean pairwise distance (MPD) metric based on the phylogenetic distance between taxa within the five dominant families for each sample. * $P < 0.05$, ** $P < 0.01$, *** $P < 0.001$, **** $P < 0.0001$, and ns: non-significant.

[Comment 8] The observation that AI predicts community structure in permafrost provides evidence that selection by environmental factors is also important, in addition to drift and dispersal. The community assembly model results are interesting, but it should also be noted that not all of the potentially important explanatory environmental factors were included. See comments below about redox, disturbance and plant community.

[Response] We agree with the reviewer's point. As mentioned above, we have added more introduction, results and discussion on the selection process (Page 4, lines 79-90; Pages 9-10, lines 205-222; Pages 11-12, lines 249-255). In terms of environmental factors, since it is hard to measure the redox potential *in situ*, especially in deep permafrost deposit along our large-scale sampling transect, we have thus added a discussion as suggested by this reviewer's [Comment #11] in our revised MS (detailed

modifications please see our response to [Comment #11]). For the plant community variables and human disturbance, we have retrieved the plant species richness and Normalized Difference Vegetation Index (NDVI) to represent plant community attributes, and human footprint (HF) index to indicate human disturbance. (Page 24, lines 514-533). These three indices have been employed in our partial mantel test (detailed modifications please see our responses to the [Comment #9]).

[Comment 9] L204-205, if any of the sites are subject to recent human disturbance, it would be good to document this and include it in the analysis. Likewise, was vegetation included in the analysis?

[Response] Good comment! We would like to mention that all sampling sites are located in remote and sparsely populated areas, most of which are above 4,000 meters and uninhabited. To consider the potential effects of vegetation and human disturbance on microbial communities, we have added the plant species richness, Normalized Difference Vegetation Index (NDVI), and the human footprint (HF) index in our analysis. We found that the plant species richness only displayed significant associations with microbial compositional variation in surface and subsurface layers (Fig. R4), and NDVI exhibited a significant effect on the microbial compositional variation in surface layer, but no significant effects on the subsurface and permafrost deposits (Fig. R4). These results indicated that the effects of vegetation on microbial compositional variation decreased with the soil depth. Additionally, the values of HF among our sites were lower than that of several typical cities (Fig. R5), suggesting that most of sampling sites were subjected minor human disturbance. Partial mantel test further confirmed that there were no significant effects of human disturbance on microbial communities in three soil layers (Fig. R4). We have added these corresponding results in our revised MS (Pages 9-10, lines 205-222).

Fig. R4: Correlations of the microbial community structure (Bray-Curtis distance) with environmental factors based on the partial mantel test. Edge width corresponds to the partial Mantel's r value, and the edge color denotes the statistical significance. Pairwise correlations of these variables are depicted with a color gradient reflecting Spearman's correlation coefficient. SUR, surface layer; SUB, subsurface layer; PL, permafrost layer; MAT, mean annual temperature; AI, aridity index; SR, plant species richness; NDVI, Normalized Difference Vegetation Index; HF, human footprint index; LCP1, labile carbon pool I (mainly polysaccharides); RCP, recalcitrant carbon pool.

Fig. R5 The comparison of human footprint (HF) index between our sampling sites and several typical cities (Lhasa, Golmud, Yushu, Nagqu, Madoi, Qilian, and Xining) on

the Tibetan Plateau. HF data were collected from National Tibetan Plateau Data Center (<https://doi.org/10.11922/sciencedb.933>). Higher HF value indicates more disturbance, and vice versa. Different lowercase letters indicate significant differences, which were determined by pairwise Wilcoxon test. Box plots indicate median (middle line), 25th, 75th percentile (box) and 5th and 95th percentile (whiskers) as well as outliers (single points).

[Comment 10] L302-307, It is true that Acidobacteria are slow-growing and stress-resistant, but they are plentiful in soils from all over the world, not just the Arctic and Antarctic. I suspect that many Chloroflexi in the permafrost are associated with fermentation. Either way, it would be useful to include a discussion of fermentative processes, as these co-dominate anoxic soil layers, along with the alternative electron accepting processes discussed here.

[Response] We agree with the reviewer's point that Acidobacteria are plentiful in soils from all over the world, not just the Arctic and Antarctic. Thus, we have rephrased the sentence as follows: "*Likewise, Acidobacteria made large contributions to arsenate reduction, selenate reduction, nitrate reduction (napAB), thiosulfate disproportionation and iron oxidation (Fig. 6a). Such diverse redox metabolism may enable their survival in anoxia and nutrient-poor conditions, consequently allowing them to be widely present in global soils⁵³*" (Pages17-18, lines 380-384).

In regard to Chloroflexi, these taxa are indeed reported to have many fermentative members, which play crucial roles in anaerobic carbon degradation in permafrost (Altshuler *et al.*, 2017). We also detected the genes involved in acetogenesis and acetate to acetyl-CoA on Chloroflexi genomes (Fig. R6). We have added this point in the revised MS as follows: "*The Chloroflexi, are reported to have many fermentative members, which play crucial roles in anaerobic carbon degradation in permafrost⁵¹. However, in this study, we observed only a moderate contribution of Chloroflexi to fermentation, and these taxa appeared to be important to the nitrite reduction process*

(nirKS and octR) (Fig. 6a)” (Page 17, lines 373-377).

Finally, as suggested by the reviewer, we have added more results (Fig. R6) and discussion about the fermentation in our revised MS as follows: “Given that soil conditions become more anoxic with depth, microbial anaerobic respiration and fermentation are essential pathways for the anaerobic decomposition of soil organic matter^{18,50}. The majority of identified anaerobic respiration genes, including those for nitrate reduction (*narGH*), sulfate reduction, and iron reduction, were prevalent in taxa within Actinobacteriota (as shown by the higher contribution on the metabolic weight scores in Fig. 6a). Likewise, genes encoding fermentation including pyruvate oxidation, pyruvate formate lyase, and acetogenesis were abundant in Actinobacteriota and several other taxa (e.g., Acidobacteriota and Proteobacteria) (Fig. 6a, Supplementary Fig. 8).... These various taxa, which have adapted to thrive in specific redox niches, ultimately co-dominant the biogeochemical processes in deeper soils¹⁷. Collectively, these results highlighted the diverse microbial species engaged in redox reactions and the more complicated trophic strategies for microorganisms in subsoils and permafrost deposits” (Pages 16-18, lines 358-389; Supplementary Fig. 8).

Fig. R6. The presence of genes encoding fermentation on the MAGs among three layers. **a** Surface layer (SUR). **b** Subsurface layer (SUB). **c** Permafrost layer (PL).

[Comment 11] Fig. 5b shows that permafrost and subsurface are metabolically more similar to each other than to the surface. This is evidence that redox conditions deeper in the soil drive selection. See for example Environmental Microbiology Reports 7.4 (2015): 649-657. As redox was not included in the 10 explanatory environmental variables, this should be discussed.

[Response] Good comment! We agree that redox conditions may drive selection in deeper soils. As suggested by the reviewer, we have added a discussion in our revised

MS as follows: “In spite of the detected importance of soil pH and AI, the effects of other variables on microbial communities should not be neglected. For example, Soil redox status was observed to exert a prominent effect on community composition and diversity in tundra soils¹⁷. Therefore, future studies are encouraged to include more explanatory environmental variables, such as ferric irons (Fe^{3+}) and electrical conductivity, to further advance our understanding of the underlying mechanisms for microbial communities in permafrost ecosystems” (Page 10, lines 216-222). Nevertheless, we did not include the redox conditions in our explanatory variables because it is hard to measure the redox potential *in situ*, especially in deep permafrost deposit along our sampling transect. Future studies should take more environmental variables into account to advance our understanding of determinants of microbial communities in permafrost ecosystems. Thanks for your understanding!

[Comment 12] Finally, check the grammar - there are some awkward phrases throughout.

[Response] We have carefully checked throughout the manuscript and modified the awkward phrases, and also asked a native English speaker (**Dr. Alistair Culf**) to polish language throughout the manuscript.

Overall, we are grateful for the insightful comments provided by this reviewer. These comments enabled us to have a deep and comprehensive thinking on clarifying our hypotheses and facilitating the discussion about our results. By addressing these comments, we feel that our revised manuscript has been greatly improved. Thank you!

Responses to Reviewer #3

[Comment 1] This is an impressive and important study describing the microbial communities across a large section of permafrost associated soils in the Tibetan Plateau. The study includes sampling of both the active layer as well as associated permafrost layer providing a link between the frozen environment and the environment. The results of this study are noteworthy in the scope of the data set providing such a large sampling transect in the north, an area being warmed by climate more rapidly than other places on the planet. Studies like this provide useful information on the microbial community and in this case the functional potential of that community.

[Response] We appreciate for the reviewer's excellent comments!

[Comment 2] The authors propose two logical hypotheses to frame the study of this system. They find broad support for both. The authors find support that drift likely imposed by dispersal limitations is driving the alpha and beta diversity gradients across the soil depth. While this may not be particularly novel, the geographic area sampled and the breadth of the study provides the novelty by increasing the sampling and knowledge of this understudied area. Using the functional potential based on the metagenomic sequencing, they find a potential enriched role for biogeochemical cycling found in permafrost soils. This result provides useful data for a better understanding of the impacts of climate change on global carbon dynamics as permafrost represents a large source of carbon.

[Response] Thanks for the reviewer's positive comments!

[Comment 3] I agree with the authors that three ecological processes are likely driving the community structure. However, you state that the order of importance is different for the permafrost layer compared to the surface layer. In your methods you indicate that you tested for differences among soil layers (L537) and Fig. 3C looks as though you compared soil layers within each ecological mechanism. Did you test for differences in mechanism within the layer as that appears to be the result you are

reporting? Based on the Fig.3, it looks like the contributions of dispersal limitation and drift are equally important.

[Response] Following the reviewer’s comment, we have examined the difference in importance of ecological processes within the layer. Our results showed that three ecological processes including dispersal limitation, drift (and others), and homogeneous selection mainly contributed to the microbial community assembly. Specifically, drift (and others) had the highest relative importance in shaping the microbial assemblages, followed very closely by dispersal limitation, then by homogeneous selection in the surface layer (Fig. R7a). For both subsurface and permafrost layers, dispersal limitation was identified as the highest contribution in structuring microbial communities, followed by the drift (and others) and homogeneous selection (Fig. R7a). We have added these results in our revised MS (Pages 10-11, lines 226-234; Fig. 3b-c).

Fig. R7: Summary of the relative contribution of ecological processes in driving

microbial communities based on null model analysis. **a** The relative contribution of each ecological processes driving microbial community assembly within layer. **b** The difference in the relative contribution of ecological processes among different soil layers. Different lowercase letters indicate significant differences for the ecological processes along the depth (Wilcoxon test, $P < 0.05$). Box plots indicate median (middle line), 25th, 75th percentile (box) and 5th and 95th percentile (whiskers) as well as outliers (single points). SUR, surface layer; SUB, subsurface layer; PL, permafrost layer;

[Comment 4] Fig. 3C uses the term “Drift and Others” but this doesn’t exactly align with the text where you use “ecological drift”. Using the same terminology in both would help the reader understand better.

[Response] As suggested by the reviewer, we have revised the “ecological drift” to “drift (and others)” in our revised MS (Pages 10-11, lines 228, 230, 234, 236, and 243).

[Comment 5] When discussing the MAGs found across the different soil layers, you found that average genome size decreased with increasing layer depth. While your explanation about compact/simplified genomes is consistent with this evidence, are there other explanations? I wonder what the recovery rate of MAGs was across the three layers that were sampled? Are there any potential biases in the number or quality of MAGs between the layers? Providing more details on the different layers may help support your explanation but may also provide alternatives for your observations.

[Response] Following the reviewer’s comments, we have examined the difference in sequencing depth, completeness, contamination, and the recovery rate of the MAGs among three layers. Our results showed that there was no significant difference in the sequencing depth, completeness, and contamination among three layers, but the recovery rate of the MAGs significantly increased with the soil depth (Fig. R8). Given that the assembly and binning biases can lead to variation in the estimation of genome size (Rodríguez-Gijón *et al.*, 2022), the difference in recovery rate of MAGs could

thus induce the bias in estimation of genome size. We have incorporated the corresponding results in the revised MS as follows: “It should be noted that the difference in the recovery rate of MAGs during the binning process (Supplementary Fig. 6) may also lead to variation in the estimation of genome size⁴³, which could, potentially, induce the decreasing pattern observed in this study” (Page 15, lines 320-323)” and “There were 34, 94, and 146 MAGs in the surface, subsurface, and permafrost layers, respectively. Most of the genomes constituted, on average, $5.9 \pm 0.6\%$, $10.1 \pm 0.6\%$, and $13.5 \pm 0.9\%$ ($n = 22$, mean \pm standard error (SE)) of the total qualified reads in the metagenomes of surface, subsurface, and permafrost layers, respectively (Supplementary Fig. 6)” (Supplementary Information page 2, lines 30-34).

Fig. R8: The difference in sequencing and MAGs characteristics among three layers determined by pairwise Wilcoxon test. **a** The difference in sequencing depth among three soil layers. **b-d** The difference in completeness (**b**) and contamination (**c**), and the recovery rate (**d**) of metagenome assembled genomes among three soil layers. Different lowercase letters denote significant differences for the genome size along the depth (Wilcoxon test, $P < 0.05$). Box plots indicate median (middle line), 25th, 75th percentile

(box) and 5th and 95th percentile (whiskers) as well as outliers (single points). SUR: surface layer, SUB: subsurface layer, and PL: permafrost layer.

[Comment 6] While I understand that the plot provided in Fig. 5A is somewhat common, the information could be presented differently to enhance the reader's understanding. The phylogenetic tree is much too small to be of any use. The other information about metabolic genes is also too condensed to be helpful.

[Response] Thanks for the reviewer's thoughtful comment. To address the size issue of the phylogenetic tree, we split the original Fig. 5A into two subplots (Fig. 5a and 5b in revised MS). Of them, Fig. 5a (as shown in Fig. R9a) showed the phylogenetic distribution, completeness, and contamination of recovery MAGs among our samples, and Fig. 5b (as shown in Fig. R9b) displayed the number of genomes per metabolic pathway by bar plot.

Fig. R9: Profile about the phylogeny, taxonomy, metabolic pathways and genome size of metagenome assembled genomes (MAGs). **a** Maximum-likelihood phylogenetic tree of the 274 MAGs from GTDB, and the completeness and contamination of MAGs. **b** The number of genomes per metabolic pathway. **c** PCoA ordination analysis using the

relative abundance of MAGs and Bray-Curtis dissimilarity index. The significant difference of the MAGs among three layers is tested by the permutational multivariate analysis of variance (PERMANOVA). **d** Comparison of estimated genome size of all MAGs found in surface (SUR), subsurface (SUB), and permafrost layer (PL). *P* values were estimated with Wilcoxon test analysis. Different lowercase letters denote significant differences for the genome size along the depth (Wilcoxon test, $P < 0.05$). Box plots indicate median (middle line), 25th, 75th percentile (box) and 5th and 95th percentile (whiskers) as well as outliers (single points).

[Comment 7] The methods section is well written and adequately describes the extensive bioinformatic pipelines. I appreciate the authors' detailed descriptions that facilitate reproducibility. Thank you for making your code available on GitHub.

[Response] Thanks for the reviewer's comments! Your recognition motivates us to keep striving for excellence in reproducibility in scientific research.

[Comment 8] The introduction states (L101) that you set up 22 sites along your permafrost transect, but in the methods (L372) you state that 24 sites were sampled. Based on the data tables, it appears that the introduction is the correct value.

[Response] Sorry for the confusion. In fact, 24 sites were sampled by our research group, while two sites were discarded in this study due to the low DNA yield. **To avoid this confusion**, we have now added this statement in *Method* section of the revised MS as follows: “*At two of these sites there were problems obtaining a sufficient DNA yield during DNA extraction and so only samples from 22 sites were processed in this study*” (Page 21, lines 452-454).

[Comment 10] Can you provide a rationale for homogenizing the soil samples from the five quadrats within a site? It seems logical for the metagenomic data, but you missed out on measuring within site variation for the 16S amplicon portion of the study. Providing this information may help others understand the design better.

[Response] Very good comment! In this study, we prefer to focus on the large-scale patterns of microbial compositional and functional attributes, and expect that the soil sampling represents the average conditions at the site level (10 m × 10 m). Due to this point, homogenizing the soil samples can account for spatial heterogeneity at each site. Similar with our sampling procedure, many studies focusing on large-scale patterns of soil biodiversity have homogenized the soil samples into one composite sample within site (Delgado-Baquerizo *et al.*, 2018; Egidi *et al.*, 2019; Karimi *et al.*, 2018). We have clearly stated this point in our revised MS as follows: “*Notably, five replicates within the 10 m × 10 m plot were expected to represent the average condition at each site, and enabling regional sampling to assess large-scale patterns of microbial compositional and functional attributes. With this aim, as done in previous studies^{65,66}, the soil samples from each layer at each site were homogenized through sterile hammering under cold conditions*” (Page 22, lines 471-476).

[Comment 11] Please consider also citing #95 at L514 for a clearer explanation of the methods for calculating the specificity and occupancy. In addition, can you clarify if you are using ASVs as the unit here or are you collapsing ASVs using the assigned taxonomy into species? The text appears to use species and ASV interchangeably when they are not.

[Response] Following the reviewer’s suggestion, we have cited #95 (now is #93) at L627 in our revised MS. Moreover, as done by Gweon *et al.* (2021), we used ASV rather species here to determine the specificity and occupancy. To avoid the confusion, we have clearly stated this point in revised MS as follows: “*Furthermore, the specificity and occupancy of each ASV were calculated in each soil layer to characterize specialist ASVs^{27,93}. Specificity is operationally defined as the average abundance of a given ASV within a set of habitat samples, while occupancy is characterized as the relative frequency with which ASV occurs within the same set of habitat samples^{27,93}. ASVs with specificity and occupancy greater or equal to 0.7 were defined as specialist species (ASVs), which indicated that they were specific to a habitat and common in most sites⁹³*”

(Page 29, lines 626-633).

[Comment 12] The SRA data doesn't appear to be available yet. Please ensure this is publicly available upon publication.

[Response] We have now provided a url link: <https://dataview.ncbi.nlm.nih.gov/object/PRJNA1037019?reviewer=9bk50klukvlfjqju72ubtf5dbq> for reviewers. All data will be publicly available upon publication.

[Comment 13] L123 change ascending to increasing to avoid a misunderstanding.

[Response] We have rephrased the sentence as follows: “*Our results showed that microbial alpha diversity declined, while beta diversity (spatial variations) increased with soil depth (Fig. 1b-e)*” (Page 7, lines 143-144).

Altogether, we appreciate for the reviewer’s insightful comments, which stimulated us to think more deeply about the results interpretation, experimental method, and the word choice, and guided us to have a thorough revision on the MS. By doing so, we feel that our revised manuscript has been greatly improved and expect that the reviewer will be satisfied with the revised manuscript. Thank you!

References

- Altshuler, I., Goordial, J., & Whyte, L. G. (2017). Microbial Life in Permafrost. In R. Margesin (Ed.), *Psychrophiles: From Biodiversity to Biotechnology* (pp. 153–179). Springer International Publishing. https://doi.org/10.1007/978-3-319-57057-0_8
- Baker, D. G., & Ruschy, D. L. (1995). Calculated and measured air and soil freeze–thaw frequencies. *Journal of Applied Meteorology*, *34*(10), 2197–2205.
- Bottos, E. M., Kennedy, D. W., Romero, E. B., Fansler, S. J., Brown, J. M., Bramer, L. M., Chu, R. K., Tfaily, M. M., Jansson, J. K., & Stegen, J. C. (2018). Dispersal limitation and thermodynamic constraints govern spatial structure of permafrost

- microbial communities. *FEMS Microbiology Ecology*, *94*(8), fiy110.
<https://doi.org/10.1093/femsec/fiy110>
- Delgado-Baquerizo, M., Oliverio, A. M., Brewer, T. E., Benavent-González, A., Eldridge, D. J., Bardgett, R. D., Maestre, F. T., Singh, B. K., & Fierer, N. (2018). A global atlas of the dominant bacteria found in soil. *Science*, *359*(6373), 320–325. <https://doi.org/10.1126/science.aap9516>
- Doherty, S. J., Barbato, R. A., Grandy, A. S., Thomas, W. K., Monteux, S., Dorrepaal, E., Johansson, M., & Ernakovich, J. G. (2020). The transition from stochastic to deterministic bacterial community assembly during permafrost thaw succession. *Frontiers in Microbiology*, *11*, 596589. <https://doi.org/10.3389/fmicb.2020.596589>
- Egidi, E., Delgado-Baquerizo, M., Plett, J. M., Wang, J., Eldridge, D. J., Bardgett, R. D., Maestre, F. T., & Singh, B. K. (2019). A few Ascomycota taxa dominate soil fungal communities worldwide. *Nature Communications*, *10*, 2369. <https://doi.org/10.1038/s41467-019-10373-z>
- Ernakovich, J. G., Barbato, R. A., Rich, V. I., Schädel, C., Hewitt, R. E., Doherty, S. J., Whalen, E. D., Abbott, B. W., Barta, J., Biasi, C., Chabot, C. L., Hultman, J., Knoblauch, C., Vetter, M. C. Y. L., Leewis, M.-C., Liebner, S., Mackelprang, R., Onstott, T. C., Richter, A., ... Winkel, M. (2022). Microbiome assembly in thawing permafrost and its feedbacks to climate. *Global Change Biology*, *28*(17), 5007–5026. <https://doi.org/10.1111/gcb.16231>
- Feng, J., Wang, C., Lei, J., Yang, Y., Yan, Q., Zhou, X., Tao, X., Ning, D., Yuan, M. M., Qin, Y., Shi, Z. J., Guo, X., He, Z., Van Nostrand, J. D., Wu, L., Bracho-Garillo, R. G., Penton, C. R., Cole, J. R., Konstantinidis, K. T., ... Zhou, J. (2020). Warming-induced permafrost thaw exacerbates tundra soil carbon decomposition mediated by microbial community. *Microbiome*, *8*(1), 3. <https://doi.org/10.1186/s40168-019-0778-3>
- Geisen, S., Tveit, A. T., Clark, I. M., Richter, A., Svenning, M. M., Bonkowski, M., & Urich, T. (2015). Metatranscriptomic census of active protists in soils. *The ISME*

- Journal*, 9(10), 2178–2190. <https://doi.org/10.1038/ismej.2015.30>
- Gweon, H. S., Bowes, M. J., Moorhouse, H. L., Oliver, A. E., Bailey, M. J., Acreman, M. C., & Read, D. S. (2021). Contrasting community assembly processes structure lotic bacteria metacommunities along the river continuum. *Environmental Microbiology*, 23(1), 484–498. <https://doi.org/10.1111/1462-2920.15337>
- Hu, W., Zhang, Q., Tian, T., Li, D., Cheng, G., Mu, J., Wu, Q., Niu, F., Stegen, J. C., An, L., & Feng, H. (2015). Relative roles of deterministic and stochastic processes in driving the vertical distribution of bacterial communities in a permafrost core from the Qinghai-Tibet Plateau, China. *PloS One*, 10(12), e0145747. <https://doi.org/10.1371/journal.pone.0145747>
- Hultman, J., Waldrop, M. P., Mackelprang, R., David, M. M., McFarland, J., Blazewicz, S. J., Harden, J., Turetsky, M. R., McGuire, A. D., Shah, M. B., VerBerkmoes, N. C., Lee, L. H., Mavrommatis, K., & Jansson, J. K. (2015). Multi-omics of permafrost, active layer and thermokarst bog soil microbiomes. *Nature*, 521(7551), 208–212. <https://doi.org/10.1038/nature14238>
- Johnston, E. R., Hatt, J. K., He, Z., Wu, L., Guo, X., Luo, Y., Schuur, E. A. G., Tiedje, J. M., Zhou, J., & Konstantinidis, K. T. (2019). Responses of tundra soil microbial communities to half a decade of experimental warming at two critical depths. *Proceedings of the National Academy of Sciences*, 116(30), 15096–15105. <https://doi.org/10.1073/pnas.1901307116>
- Karimi, B., Terrat, S., Dequiedt, S., Saby, N. P. A., Horrigue, W., Lelièvre, M., Nowak, V., Jolivet, C., Arrouays, D., Wincker, P., Cruaud, C., Bispo, A., Maron, P.-A., Bouré, N. C. P., & Ranjard, L. (2018). Biogeography of soil bacteria and archaea across France. *Science Advances*, 4(7), eaat1808. <https://doi.org/10.1126/sciadv.aat1808>
- Mackelprang, R., Burkert, A., Haw, M., Mahendrarajah, T., Conaway, C. H., Douglas, T. A., & Waldrop, M. P. (2017). Microbial survival strategies in ancient permafrost: Insights from metagenomics. *The ISME Journal*, 11(10), 2305–2318.

<https://doi.org/10.1038/ismej.2017.93>

- Mackelprang, R., Waldrop, M. P., DeAngelis, K. M., David, M. M., Chavarria, K. L., Blazewicz, S. J., Rubin, E. M., & Jansson, J. K. (2011). Metagenomic analysis of a permafrost microbial community reveals a rapid response to thaw. *Nature*, *480*(7377), 368-371. <https://doi.org/10.1038/nature10576>
- Mao, C., Kou, D., Chen, L., Qin, S., Zhang, D., Peng, Y., & Yang, Y. (2020). Permafrost nitrogen status and its determinants on the Tibetan Plateau. *Global Change Biology*, *26*(9), 5290–5302. <https://doi.org/10.1111/gcb.15205>
- McCalley, C. K., Woodcroft, B. J., Hodgkins, S. B., Wehr, R. A., Kim, E.-H., Mondav, R., Crill, P. M., Chanton, J. P., Rich, V. I., Tyson, G. W., & Saleska, S. R. (2014). Methane dynamics regulated by microbial community response to permafrost thaw. *Nature*, *514*(7523), 478-481. <https://doi.org/10.1038/nature13798>
- Mondav, R., McCalley, C. K., Hodgkins, S. B., Froelking, S., Saleska, S. R., Rich, V. I., Chanton, J. P., & Crill, P. M. (2017). Microbial network, phylogenetic diversity and community membership in the active layer across a permafrost thaw gradient. *Environmental Microbiology*, *19*(8), 3201–3218. <https://doi.org/10.1111/1462-2920.13809>
- Mondav, R., Woodcroft, B. J., Kim, E.-H., McCalley, C. K., Hodgkins, S. B., Crill, P. M., Chanton, J., Hurst, G. B., VerBerkmoes, N. C., Saleska, S. R., Hugenholtz, P., Rich, V. I., & Tyson, G. W. (2014). Discovery of a novel methanogen prevalent in thawing permafrost. *Nature Communications*, *5*(1), 3212. <https://doi.org/10.1038/ncomms4212>
- Müller, O., Bang-Andreasen, T., White III, R. A., Elberling, B., Taş, N., Kneafsey, T., Jansson, J. K., & Øvreås, L. (2018). Disentangling the complexity of permafrost soil by using high resolution profiling of microbial community composition, key functions and respiration rates. *Environmental Microbiology*, *20*(12), 4328–4342. <https://doi.org/10.1111/1462-2920.14348>
- Qin, S., Kou, D., Mao, C., Chen, Y., Chen, L., & Yang, Y. (2021). Temperature sensitivity of permafrost carbon release mediated by mineral and microbial

- properties. *Science Advances*, 7(32), eabe3596.
<https://doi.org/10.1126/sciadv.abe3596>
- Rodríguez-Gijón, A., Nuy, J. K., Mehrshad, M., Buck, M., Schulz, F., Woyke, T., & Garcia, S. L. (2022). A genomic perspective across Earth's microbiomes reveals that genome size in archaea and bacteria is linked to ecosystem type and trophic strategy. *Frontiers in Microbiology*, 12, 761869.
<https://www.frontiersin.org/journals/microbiology/articles/10.3389/fmicb.2021.761869>
- Singleton, C. M., McCalley, C. K., Woodcroft, B. J., Boyd, J. A., Evans, P. N., Hodgkins, S. B., Chanton, J. P., Frolking, S., Crill, P. M., Saleska, S. R., Rich, V. I., & Tyson, G. W. (2018). Methanotrophy across a natural permafrost thaw environment. *The ISME Journal*, 12(10), 2544–2558.
<https://doi.org/10.1038/s41396-018-0065-5>
- Song, Y., Chen, L., Kang, L., Yang, G., Qin, S., Zhang, Q., Mao, C., Kou, D., Fang, K., Feng, X., & Yang, Y. (2021). Methanogenic community, CH₄ production potential and its determinants in the active layer and permafrost deposits on the Tibetan Plateau. *Environmental Science & Technology*, 55(16), 11412–11423.
<https://doi.org/10.1021/acs.est.0c07267>
- Tang, X., Zhang, M., Fang, Z., Yang, Q., Zhang, W., Zhou, J., Zhao, B., Fan, T., Wang, C., Zhang, C., Xia, Y., & Zheng, Y. (2023). Changing microbiome community structure and functional potential during permafrost thawing on the Tibetan Plateau. *FEMS Microbiology Ecology*, 99(11), fiad117.
<https://doi.org/10.1093/femsec/fiad117>
- Taş, N., Prestat, E., McFarland, J. W., Wickland, K. P., Knight, R., Berhe, A. A., Jorgenson, T., Waldrop, M. P., & Jansson, J. K. (2014). Impact of fire on active layer and permafrost microbial communities and metagenomes in an upland Alaskan boreal forest. *The ISME Journal*, 8(9), 1904–1919.
<https://doi.org/10.1038/ismej.2014.36>
- Taş, N., Prestat, E., Wang, S., Wu, Y., Ulrich, C., Kneafsey, T., Tringe, S. G., Torn, M.

- S., Hubbard, S. S., & Jansson, J. K. (2018). Landscape topography structures the soil microbiome in arctic polygonal tundra. *Nature Communications*, *9*(1), 777. <https://doi.org/10.1038/s41467-018-03089-z>
- Tveit, A., Schwacke, R., Svenning, M. M., & Urich, T. (2013). Organic carbon transformations in high-Arctic peat soils: Key functions and microorganisms. *The ISME Journal*, *7*(2), 299–311. <https://doi.org/10.1038/ismej.2012.99>
- Tveit, A. T., Urich, T., Frenzel, P., & Svenning, M. M. (2015). Metabolic and trophic interactions modulate methane production by Arctic peat microbiota in response to warming. *Proceedings of the National Academy of Sciences*, *112*(19), E2507–E2516. <https://doi.org/10.1073/pnas.1420797112>
- Vishnivetskaya, T. A., Almatari, A. L., Spirina, E. V., Wu, X., Williams, D. E., Pfiffner, S. M., & Rivkina, E. M. (2020). Insights into community of photosynthetic microorganisms from permafrost. *FEMS Microbiology Ecology*, *96*(12), fiae229. <https://doi.org/10.1093/femsec/fiae229>
- Waldrop, M. P., Chabot, C. L., Liebner, S., Holm, S., Snyder, M. W., Dillon, M., Dudgeon, S. R., Douglas, T. A., Lewis, M.-C., Walter Anthony, K. M., McFarland, J. W., Arp, C. D., Bondurant, A. C., Taş, N., & Mackelprang, R. (2023). Permafrost microbial communities and functional genes are structured by latitudinal and soil geochemical gradients. *The ISME Journal*, *17*, 1224–1235. <https://doi.org/10.1038/s41396-023-01429-6>
- Webb, C. O. (2000). Exploring the phylogenetic structure of ecological communities: an example for rain forest trees. *The American Naturalist*, *156*(2), 145–155. <https://doi.org/10.1086/303378>
- Wei, D., Qi, Y., Ma, Y., Wang, X., Ma, W., Gao, T., Huang, L., Zhao, H., Zhang, J., & Wang, X. (2021). Plant uptake of CO₂ outpaces losses from permafrost and plant respiration on the Tibetan Plateau. *Proceedings of the National Academy of Sciences*, *118*(33), e2015283118. <https://doi.org/10.1073/pnas.2015283118>
- Woodcroft, B. J., Singleton, C. M., Boyd, J. A., Evans, P. N., Emerson, J. B., Zayed, A. A. F., Hoelzle, R. D., Lamberton, T. O., McCalley, C. K., Hodgkins, S. B.,

- Wilson, R. M., Purvine, S. O., Nicora, C. D., Li, C., Frohling, S., Chanton, J. P., Crill, P. M., Saleska, S. R., Rich, V. I., & Tyson, G. W. (2018). Genome-centric view of carbon processing in thawing permafrost. *Nature*, *560*(7716), 49-54. <https://doi.org/10.1038/s41586-018-0338-1>
- Wu, L., Yang, F., Feng, J., Tao, X., Qi, Q., Wang, C., Schuur, E. A. G., Bracho, R., Huang, Y., Cole, J. R., Tiedje, J. M., & Zhou, J. (2022). Permafrost thaw with warming reduces microbial metabolic capacities in subsurface soils. *Molecular Ecology*, *31*(5), 1403–1415. <https://doi.org/10.1111/mec.16319>
- Wu, M.-H., Xue, K., Wei, P.-J., Jia, Y.-L., Zhang, Y., & Chen, S.-Y. (2022). Soil microbial distribution and assembly are related to vegetation biomass in the alpine permafrost regions of the Qinghai-Tibet Plateau. *Science of The Total Environment*, *834*, 155259. <https://doi.org/10.1016/j.scitotenv.2022.155259>
- Wu, X., Almatari, A. L., Cyr, W. A., Williams, D. E., Pfiffner, S. M., Rivkina, E. M., Lloyd, K. G., & Vishnivetskaya, T. A. (2023). Microbial life in 25-m-deep boreholes in ancient permafrost illuminated by metagenomics. *Environmental Microbiome*, *18*(1), 1-19. <https://doi.org/10.1186/s40793-023-00487-9>
- Wu, X., Chauhan, A., Layton, A. C., Lau Vetter, M. C. Y., Stackhouse, B. T., Williams, D. E., Whyte, L., Pfiffner, S. M., Onstott, T. C., & Vishnivetskaya, T. A. (2021). Comparative metagenomics of the active layer and permafrost from low-carbon soil in the Canadian High Arctic. *Environmental Science & Technology*, *55*(18), 12683–12693. <https://doi.org/10.1021/acs.est.1c00802>
- Xue, K., M. Yuan, M., J. Shi, Z., Qin, Y., Deng, Y., Cheng, L., Wu, L., He, Z., Van Nostrand, J. D., Bracho, R., Natali, S., Schuur, E. A. G., Luo, C., Konstantinidis, K. T., Wang, Q., Cole, J. R., Tiedje, J. M., Luo, Y., & Zhou, J. (2016). Tundra soil carbon is vulnerable to rapid microbial decomposition under climate warming. *Nature Climate Change*, *6*(6), 595–600. <https://doi.org/10.1038/nclimate2940>
- Yang, K., Yang, K., & Su, B. (2019). Time-lapse observation dataset of soil temperature and humidity on the Tibetan Plateau (2008-2016). National Tibetan Plateau

Data Center. <https://doi.org/10.11888/Soil.tpd.270110>

Yergeau, E., Hogues, H., Whyte, L. G., & Greer, C. W. (2010). The functional potential of high Arctic permafrost revealed by metagenomic sequencing, qPCR and microarray analyses. *The ISME Journal*, 4(9), 1206–1214.

REVIEWERS' COMMENTS

Reviewer #2 (Remarks to the Author):

The authors have thoroughly and thoughtfully addressed my earlier comments and concerns. I believe it is acceptable for publication.

Reviewer #3 (Remarks to the Author):

All of my previous concerns have been addressed in the revised manuscript. I would like to thank the authors for the revisions and clarifications.